# Accounting for population structure and data quality in demographic inference with linkage disequilibrium methods

Enrique Santiago ®[1] ✉, Carlos Köpke ®[2] & Armando Caballero[3]

Linkage disequilibrium methods for demographic inference usually rely on panmictic population models. However, the structure of natural populations is generally complex and the quality of the genotyping data is often suboptimal. We present two software tools that implement theoretical developments to estimate the effective population size ($N_e$): *GONE2*, for inferring recent changes in $N_e$ when a genetic map is available, and *currentNe2*, which estimates contemporary $N_e$ even in the absence of genetic maps. These tools operate on SNP data from a single sample of individuals, and provide insights into population structure, including the $F_{ST}$ index, migration rate, and subpopulation number. *GONE2* can also handle haploid data, genotyping errors, and low sequencing depth data. Results from simulations and laboratory populations of *Drosophila melanogaster* validated the tools in different demographic scenarios, and analysis were extended to populations of several species. These results highlight that ignoring population subdivision often leads to $N_e$ underestimation.

The concept of effective population size ($N_e$)[1] is fundamental to population genetics, quantifying the extent of genetic drift, inbreeding, and adaptive potential. Understanding recent variation in $N_e$ is relevant for assessing conservation status and guiding management strategies[2,3]. Traditional methods for estimating $N_e$[4–7] provide limited resolution of recent demographic change. Coalescent methods, which typically rely on mutation rates to determine the time of occurrence of demographic events in the past, are more appropriate for estimating $N_e$ in ancient times than for recent demography[8,9]. Methods based on the distribution of identity-by-descent (IBD) lengths across the genome[10–13] require high-quality phased genotypes and dense marker sets, limiting their utility in non-model systems.

Linkage disequilibrium (LD)-based methods have emerged as powerful tools for inferring demographic history from single-sample genomic data, even with moderate data quality[14]. These methods can be traced back to the seminal work by Hill and Robertson[15], who established the relationship between recombination rates and disequilibrium decay in finite populations. As this decay varies significantly with recombination rates, comparing observed LD between loci at different distances provides insights into population dynamics

in the recent past[10,16–18]. An advantage of LD methods is the possibility to infer $N_e$ from a small number of unphased genotypes, even in the absence of a genetic map[19], provided that the total genetic size of the genome is known. This makes LD methods particularly valuable for the study of non-model species where comprehensive genetic resources may be lacking. Several software tools have been developed to use LD data for demographic inference, including *NeEstimator*[20], *GONE*[18], *currentNe*[21], and *HapNe-LD*[13].

A persistent limitation of existing LD tools is their reliance on panmictic population models, which can introduce substantial biases in $N_e$ estimates when applied to structured populations[22–25]. This shortcoming has critical implications for conservation genetics, where the magnitude of $N_e$ is used as an indicator of long-term population persistence[26], and understanding genetic connectivity between subpopulations is critical for developing effective conservation strategies. The work of Ragsdale and Gravel[27] approached this problem by using transition matrices of two locus moments but, while theoretically comprehensive, the method is computationally intractable when applied to genome-scale datasets of randomly sampled individuals.

[1]Departamento de Biología Funcional, Facultad de Biología, Universidad de Oviedo, Oviedo, Spain. [2]Plasma Labs Enterprises SL, Oviedo, Spain. [3]Centro de Investigación Mariña, Universidade de Vigo, Facultade de Bioloxía, Vigo, Spain. ✉e-mail: esr@uniovi.es

To address this challenge, we present theoretical developments implemented in the latest releases of *GONE2* and *currentNe2*, that allow the effective size, the genetic differentiation index $F_{ST}$, the migration rate, and the number of subpopulations to be inferred from a single sample of individuals taken at random from a population. The method combines the information on the LD of pairs of sites in different chromosomes, the LD between weakly linked sites, and the average inbreeding coefficient. This approach provides deeper insight into the recent demography in geographically structured populations. In addition, *GONE2* includes new features such as the analysis of haploid data, and the handling of genotyping error rates and low sequencing depth, which are typical problems encountered particularly when analyzing ancient DNA[28]. Both software releases are also more compact and significantly faster than the previous versions, making them easier to use in a single command-line format and enabling the analysis of larger samples.

## Results

### Accounting for population structure

Linkage disequilibrium ($\delta^2$) is defined by the ratio of expectations of the squared covariances ($D^2$) and the products of variances ($W$) of pairs of loci[29]. An island model[30] of $s$ equal-sized subpopulations and reciprocal migration rate $m$ is assumed across the theoretical developments. For this model, the LD can be partitioned into the within-subpopulations component $\delta_w^2$, the between-subpopulations component $\delta_b^2$ and the between-within component $\delta_{bw}^2$ (see Section 2 of the Supplementary Information):

$$\delta^2 = \delta_w^2 + \delta_b^2 + 2 \cdot \delta_{bw}^2 \qquad (1)$$

In a metapopulation at migration-drift equilibrium, the expectations of the terms in the equation becomes:

$$E\left[\delta_w^2\right] = (1 - F_{ST})^2 \cdot \frac{1 + c^2}{2N_T\left(1 - (1 - c)^2\right) + 2.2(1 - c)^2}$$

where $c$ is the recombination rate between loci, and $F_{ST}$ is Wright's[30] differentiation index. This formulation scales panmictic LD expectations by $(1 - F_{ST})^2$, where $N_T$ represents the summed effective sizes of all subpopulations. In the context of metapopulations, this concept differs from the effective metapopulation size $N_e$, which refers to the effective population size of drift for the entire metapopulation[31]. At migration-drift equilibrium, these two concepts are linked by the equation $N_e = N_T/(1 - F_{ST})$ [32]. The between deme component follows:

$$E\left[\delta_b^2\right] \approx \frac{F_{ST}^2}{s - 1}$$

where $s$ is the number of subpopulations. The between-within component approximates:

$$E\left[\delta_{bw}^2\right] \approx \frac{\left(\frac{s}{s-1}\right)^2 \delta_b^2 \cdot m}{1 - \left(1 - \frac{s}{s-1}m\right)^2 \cdot (1 - c)}$$

(see Section 2.2 of the Supplementary Information for a more accurate equation for closely linked sites).

Empirical validation with simulations of metapopulations with two and four subpopulations (Supplementary Tables 1 and 2) confirms this partitioning across recombination regimes, population sizes and migration rates. Supplementary Fig. 1 illustrates the theoretical LD partitioning over a range of recombination rates in a metapopulation with two subpopulations.

If the $\delta^2$ value for a given recombination rate $c$ is known, four explicit unknowns remain in Eq. (1): the total metapopulation size ($N_T$),

the migration rate ($m$), the genetic differentiation index ($F_{ST}$), and the number of subpopulations ($s$). These variables are linked by the general equation for $F_{ST}$ of Takahata[33] (see Eq. A14 in the Supplementary Information), which reduces the number of unknowns to three. These can be solved using at least three independent parameters measured from the sample. After testing different combinations, we chose the measurements of $\delta^2$ for unlinked sites ($c = 0.5$; on different chromosomes) and for weakly linked sites ($c > 0.05$; on the same chromosome), and the inbreeding coefficient observed in the sample. The latter serves as an approximation for $F_{ST}$ under the assumption of panmixia within subpopulations, and its combination with LD metrics increases the stability of parameter estimates. Santiago et al.[21] showed that most of the linkage disequilibrium observed at high or intermediate recombination rates is due to drift in very recent generations. Assuming no drastic recent demographic changes, it is reasonable to propose that similar values of $N_e$ (the current effective size), $m$, s, and $F_{ST}$ are involved in the equation for the expected $\delta^2$ between sites on different chromosomes (area B in Supplementary Fig. 1) and in a second equation for the expected $\delta^2$ between distant sites on the same chromosome (area A in Supplementary Fig. 1). Consequently, these two equations, together with Takahata's general equation[33] for $F_{ST}$ and the observed inbreeding value, jointly estimate the four unknowns through a numerical optimization that minimizes the sum of squares of the differences between observed and predicted $\delta^2$ for the two sets of sites. This procedure is implemented in *GONE2*, which always requires a genetic map, and in *currentNe2*, which operates without restrictions on genetic distances within chromosomes when no map is available.

*GONE2*, like its predecessor, uses a hidden Markov process to estimate the historical $N_e$ series under the assumption of a single panmictic population (the default option). It compares the observed LD across recombination bins with the predicted LD derived from proposed demographic histories, ultimately identifying the history that best fits the LD observations across all bins. However, to account for metapopulations in the Markov process, temporal migration rates must be included as hidden parameters, complicating the search for a solution. To address this challenge, *GONE2* uses an approach inspired by Tenesa et al. [34], which assumes that the LD between loci with recombination rate $c$ reflects the $N_e$ of $1/(2c)$ generations ago, assuming linear changes in $N_e$[10]. After substituting the estimates of $m$, s, and $F_{ST}$, calculated as described above, and the observed $\delta^2$ value for a given recombination bin into Eq. (1), this method generates a $N_e$ estimate for the $c$ value of the bin. LD of bins with lower $c$ values correspond to $N_e$ estimates deeper in the past. While this approach is computationally efficient in complex metapopulation scenarios, it provides only an approximate demographic trend as it does not take into account that the LD corresponding to a particular bin is influenced by the entire demographic history of the population, as *GONE2* does when panmixia is assumed.

### Correction for base-calling errors

In a two-locus biallelic model, genotyping errors at one locus replace one allele with the other, inducing a new covariance of opposite sign. Let $D_{i,j}$ be the true covariance between loci $i$ and $j$ in the population, and $\varepsilon$ be the probability of genotyping error per base read. The expected covariance after genotyping is the sum of the remaining original covariance and the new covariance:

$$D_{i,j}' = D_{i,j} \cdot [1 - 2\varepsilon(1 - \varepsilon)] - D_{i,j} \cdot 2\varepsilon(1 - \varepsilon) = D_{i,j} \cdot [1 - 4\varepsilon(1 - \varepsilon)]$$

The pre-genotyping variance at a locus $i$ with allele frequencies $p_i$ and $(1 - p_i)$ is $V_i = p_i(1 - p_i)$. Therefore, the expected post-genotyping

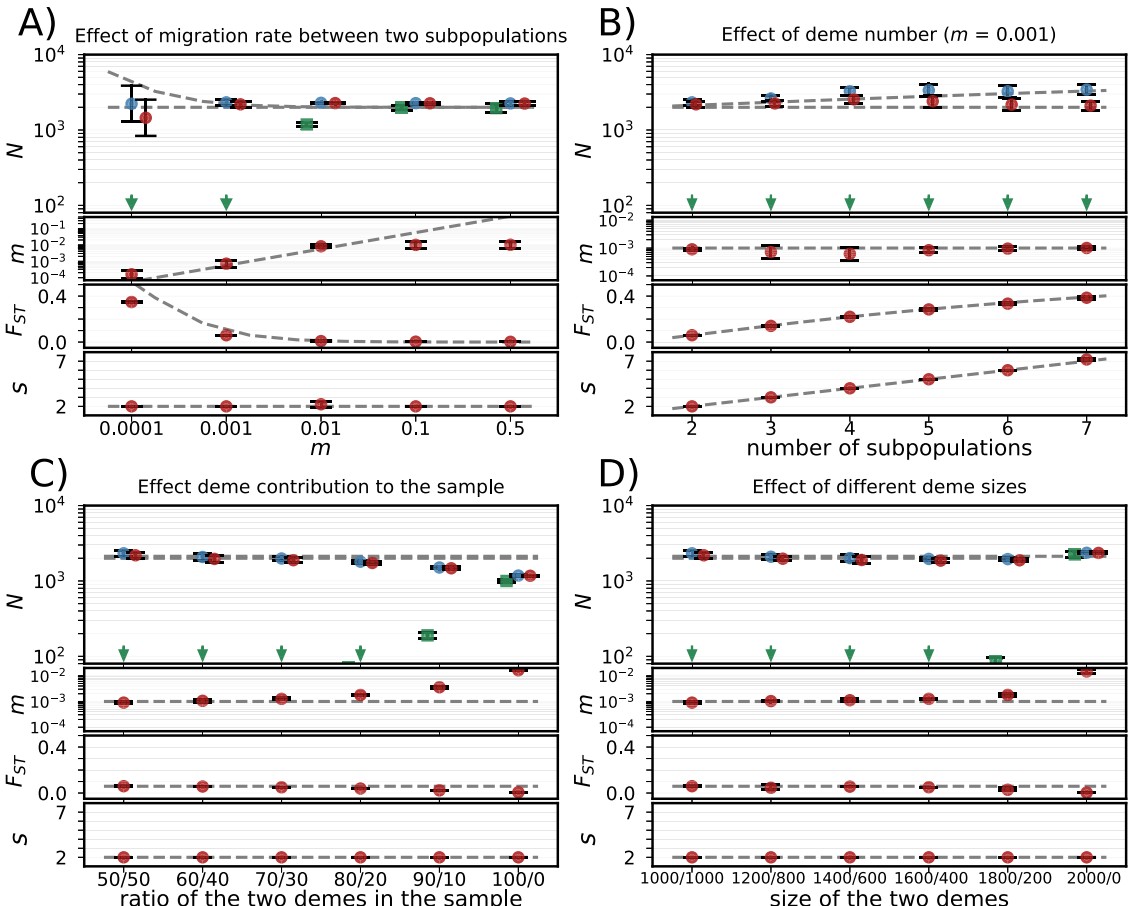

**Fig. 1 | currentNe2 estimates in different simulated scenarios.** The top plot in each panel shows estimates of contemporary effective population size ($N_e$) for the entire metapopulation (blue dots), the sum of the effective sizes of the subpopulations ($N_T$, red dots), and $N_e$ assuming panmixia (green squares; green arrows indicate values below the scale limit). Note that the labels '$N$' on the x-axes here refer generically to the three estimates. The three lower plots show estimates for migration rates ($m$), genetic differentiation indices ($F_{ST}$), and the number of subpopulations ($s$). All estimates were calculated using the *currentNe2* under the assumption of population subdivision (option '-x'), except for $N_e$ under panmixia. Each estimate represents the geometric mean of ten simulated metapopulations, with 95% confidence intervals. True metapopulation values are indicated by grey dashed lines ($N_e$ and $N_T$ lines often overlap, with $N_e$ typically higher). **A** Effect of varying migration rates in a metapopulation with two subpopulations of 1000 individuals each. **B** Effect of increasing the number of equally sized subpopulations while maintaining a total metapopulation size of 2000 individuals ($m = 0.001$). **C** Effect of unequal sampling proportions between two subpopulations of 1000 individuals each ($m = 0.001$). **D** Effect of unequal subpopulation sizes in a metapopulation with two subpopulations totalling 2000 individuals ($m = 0.001$). All estimates are based on random samples of 100 individuals from the entire metapopulation, except in (**C**), where sampling proportions vary. Source data are provided as a Source Data file.

variance at locus $i$ is

$$V_i' = V_i + \varepsilon\left[1 - 4V_i\right]$$

These equations give the deterministic expectations for $D_{i,j}'^2$ and for $W_{i,j}' = V_i'V_j'$ in the population. Assuming sampling with replacement over these expectations, the finite sampling corrections are derived (see Section 1.10 of the Supplementary Information). Even with the above corrections, the presence of genotyping errors adds noise to the estimates of LD, and represents a major handicap in estimating the small values of LD for unlinked and weakly linked loci. Consequently, the consideration of genotyping errors has been excluded from *currentNe2* and also from the calculation of LD by *GONE2* for subdivided populations, both assuming high quality of genotyping.

**Non-parametric solution for low sequencing depth**

To address the challenge of low sequencing depth in diploid organisms, which can lead to increased observed homozygosity and biased LD measurements, *GONE2* uses the pseudohaploid theory developed in Section 1.9 of the Supplementary Information. This approach

assumes no intrinsic allele preference in the sequencing method at heterozygous sites. The software generates pseudohaploid genomes for each diploid individual in the sample by randomly selecting one allele at each observed heterozygous site[35]. *GONE2* then estimates the historical $N_e$ using these pseudohaploid samples. To increase accuracy, this process is repeated fifty times, with the partial estimates averaged to produce a final, corrected estimate of historical $N_e$. This method effectively accounts for variable sequencing depth across the genome, allowing for differences in sequencing depth both between individuals and within genomes. For the same reasons as for genotyping error, the implementation of the correction for low sequencing depth is only included in *GONE2* when panmictic populations are assumed.

**Checking predictions with simulated and empirical data**

Figure 1 shows *currentNe2* estimates of key metapopulation parameters under different island model scenarios, including deviations from equal deme sizes and sampling. The metapopulation analysis, invoked with the '-x' option, generally provides accurate $N_e$ estimates under all scenarios. However, migration rate estimates become less reliable at low deme differentiation, while $F_{ST}$ is underestimated at very

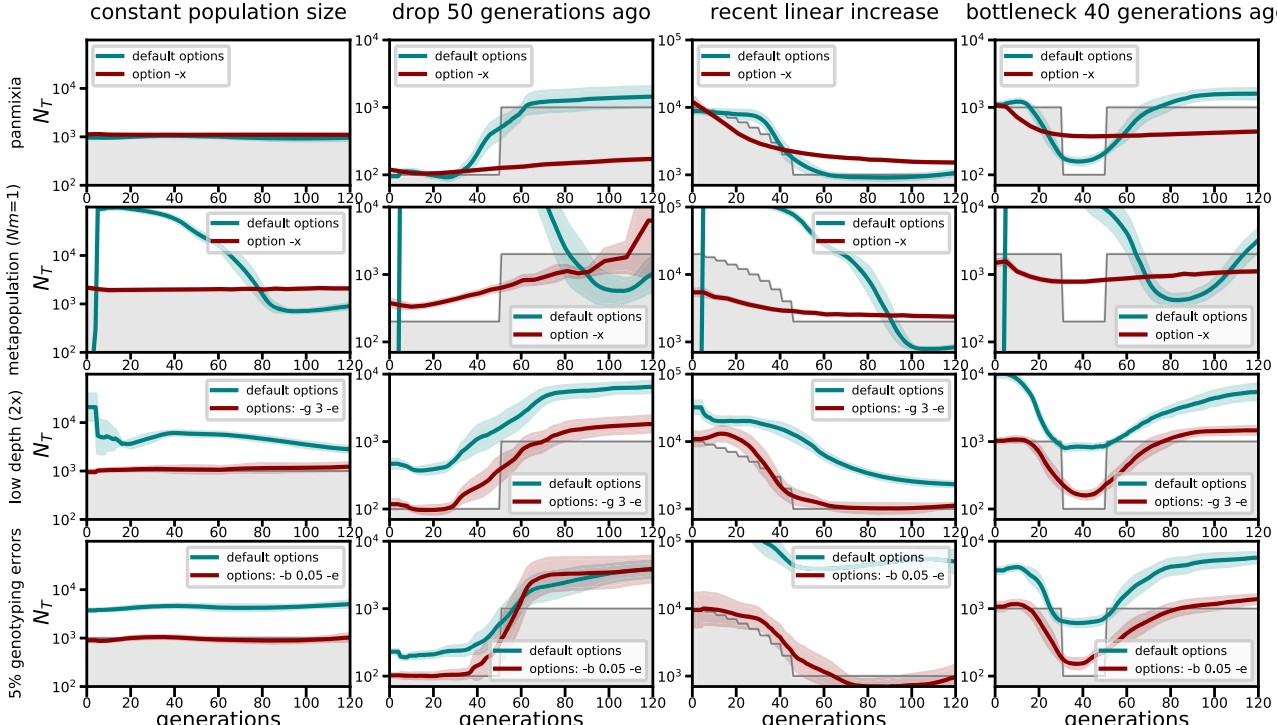

**Fig. 2 | GONE2 estimates of the historical $N_e$ in different simulated scenarios.** Four different profiles of demographic change are represented in the columns by the grey-shaded areas: constant population size, recent decline, recent expansion, and bottleneck. The scenarios are also arranged in rows in four different situations. First row: Estimates from panmictic populations using the default option of the program (green lines) and the '-x' option, which incorrectly assumes that the population is divided in two subpopulations (red lines). Second row: Estimates from metapopulations composed of two subpopulations of equal size $N$ and migration rate $m = 0.001$ per generation. The analyses consider either a metapopulation divided into two subpopulations (option '-x', red lines) or the incorrect assumption of a single panmictic population (default program options, green lines). Third row: Estimates from low depth of sequencing data (average depth 2x) of panmictic populations. Green lines show estimates not corrected for low sequencing depth (default program options), while red lines show estimates corrected for low sequencing depth (option '-g 3' to be used for any sequencing depth). Fourth row: Estimates from sequencing data with 5% base calling errors. Green lines represent unadjusted estimates not corrected for errors (program default option), while red lines represent estimates corrected for the true 5% error rate (option '-b 0.05'). Simulation results are the geometric mean of 20 replicates. The shaded areas around the estimate lines represent the 95% interval of the distribution of the estimates. Source data are provided as a Source Data file.

low migration rates ($Nm \ll 1$, where $N$ is the subpopulation size). When sampling from a single subpopulation, both analytical approaches (default option and '-x' option) yield similar estimates of the effective size of the subpopulation (Fig. 1C). It is notable that, when population subdivision is assumed, the estimates of $N_e$ of the total metapopulation remain close to the true values regardless of differences in subpopulation sizes, even under extreme asymmetries (Fig. 1D).

Assuming a single panmictic population (the default option) gives the same estimates as the previous version of *currentNe*[21]: When analyzing a random sample from the entire metapopulation, this leads to large underestimates of $N_e$ unless migration rates are so high ($Nm \gg 1$) that the metapopulation effectively behaves as a single panmictic population. These results suggest that comparing $N_e$ estimates assuming metapopulation structure (the '-x' option) with those assuming panmixia (the default option) can help to identify population structure: similar $N_e$ estimates using both options indicate a panmictic population, while lower $N_e$ estimates using the default option indicate subpopulation differentiation. In the latter case, the '-x' option should be used to estimate $N_e$. These results hold even with significant departures from the island model, such as unequal deme sizes and asymmetric migration (Supplementary Fig. 2). Estimates remain robust for the continent-island and the stepping stone models, except for less accurate subpopulation number estimates in the latter.

Supplementary Table 3 shows that for panmictic populations, $N_e$ estimates with and without the '-x' option have overlapping 90% confidence intervals, indicating no significant difference. However, for

highly structured populations, these estimates diverge significantly. The threshold for detecting metapopulation structure appears to be around $Nm = 10$ (Supplementary Table 4), highlighting the sensitivity of LD between unlinked and weakly linked sites to population differentiation.

Figure 2 shows *GONE2* estimates for different simulated scenarios. For panmictic populations (row 1), both the default and '-x' options give similar estimates for recent generations, but the '-x' option fails to accurately capture older demographic changes. This suggests that the default option, which gives the same results as the previous version of *GONE*[18], is preferable for panmictic populations or weakly differentiated subpopulations. In metapopulation scenarios (row 2), the default option incorrectly indicates a recent decline in size, while the '-x' option corrects this bias but only captures general demographic trends. Additional comparisons with *HapNe-LD*[13] estimates (Supplementary Fig. 3) show that *GONE2* with the '-x' option performs better in terms of accuracy and trend detection under constant and variable demographic scenarios.

Table 1 compares $N_e$ estimates with a wide range of sample sizes across different animal species, generally indicating subpopulation differentiation. This was to be expected for multi-breed samples, with the smallest difference in the sample mix of the two closely related breeds, Gordon and English Setters. To varying degrees, the gap between the two $N_e$ estimates, assuming panmixia (default) and assuming structure (option '-x ') is clear in all natural populations, except for the two salmon samples, for which the 90% CIs assuming

**Table 1 | *currentNe2* estimates with samples from different species**

| Species/Population/Sample | n (v) | Estimates Assuming Panmixia | | Estimates Assuming Structure | | | | | |
|---|---|---|---|---|---|---|---|---|---|
| | | $N_e$ | CI | $N_e$ | CI | s | m | $F_{ST}$ |
| Boar (*S. scrofa*) wild boars from the island of Sardinia[53] | 99 (33) | 55 | 51–60 | 888 | 719–1096 | 13 | 0.01920 | 0.16243 |
| Cattle (*B. taurus*) 134 cattle breeds sampled worldwide[54] | 1543 (9) | 12 | 12–12 | 2241 | 2214–2268 | 5 | 0.00170 | 0.21045 |
| Deer (*O.virginianus*) sampled throughout USA and Canada[55] | 43 (100) | 37 | 35–39 | 64,825 | 24,897–168,786 | 6 | 0.00011 | 0.15176 |
| Dog (*Canis lupus familiaris*) Gordon and English Setters[56] | 261 (109) | 42 | 42–43 | 94 | 92–97 | 2 | 0.03054 | 0.04345 |
| Flycatcher (*F. albicollis*) sampled in the island of Öland[57] | 864 (19) | 830 | 823–838 | 6732 | 6376–7108 | 2 | 0.00110 | 0.01684 |
| Goat (*C. hircus*) 15 breeds from Italy and 26 from Pakistan[58] | 929 (29) | 21 | 21–21 | 40,421 | 35,293–46,294 | 6 | 0.00014 | 0.17381 |
| Gorilla (*G.gorilla* - West Lowland), mostly from Cameroon[59] | 14 (99) | 73 | 46–117 | 2394 | 456–12,568 | 7 | 0.00551 | 0.09748 |
| Horse (*E. caballus*) Egyptian Arabian breed[60] | 36 (15) | 14 | 12–16 | 21 | 18–23 | 2 | 0.12218 | 0.04927 |
| Orca (*O. orca*) sampled in the eastern Canadian High Arctic[61] | 16 (100) | 42 | 31–57 | 1341 | 434–4138 | 2 | 0.00214 | 0.04360 |
| Pig (*S. domesticus*) 6 breeds from Italy and 5 worldwide[53] | 105 (37) | 18 | 17–19 | 207 | 186–229 | 10 | 0.03904 | 0.25113 |
| Salmon (*S. salar*) from Baddoch, tributary of the river Dee[a] | 16 (158) | 227 | 128–405 | 507 | 233–1102 | 2 | 0.01317 | 0.01874 |
| Salmon (*S. salar*) from Girnock, tributary of the river Dee[a] | 16 (152) | 221 | 125–394 | 491 | 228–1060 | 2 | 0.00969 | 0.02625 |
| Seabass (*D. labrax*) from the FMD breeding programme[49] | 65 (3) | 57 | 51–63 | 55 | 49–61 | 2 | 0.06608 | 0.03447 |
| Seabream (*S. aurata*) Andromeda breeding programme[49] | 117 (9) | 80 | 73–87 | 65 | 60–70 | 2 | 0.07031 | 0.02749 |
| Seabream (*S. aurata*) from the FMD breeding programme[49] | 107 (22) | 65 | 60–71 | 28 | 26–29 | 2 | 0.49994 | 0.00902 |
| Sheep (*O. aries*) 8 breeds from Algeria[62] | 46 (43) | 175 | 138–222 | 1572 | 916–2697 | 9 | 0.01638 | 0.06906 |
| Turbot (*S. maximus*) CETGA experimental population[49] | 46 (16) | 62 | 54–71 | 99 | 85–350 | 2 | 0.02065 | 0.06114 |

Estimates of contemporary effective population sizes ($N_e$), 90% confidence intervals (CI), number of subpopulations (s), migration rates (m) and fixation index ($F_{ST}$) obtained by *currentNe2* from empirical animal data (n: number of individuals in the sample; v ×10³: number of SNPs). Estimates assume either geographic structure (option '-x') or panmixia (default option).
[a]Data provided by John Taggart (Institute of Aquaculture, University of Stirling) and available at https://github.com/esrud/currentNe2/tree/main/examples; Assumed rates of recombination of 1 cM/Mb for all species except for Gorilla (1.18), Seabass (2.38), Seabream (2.13), and Turbot (2.47).

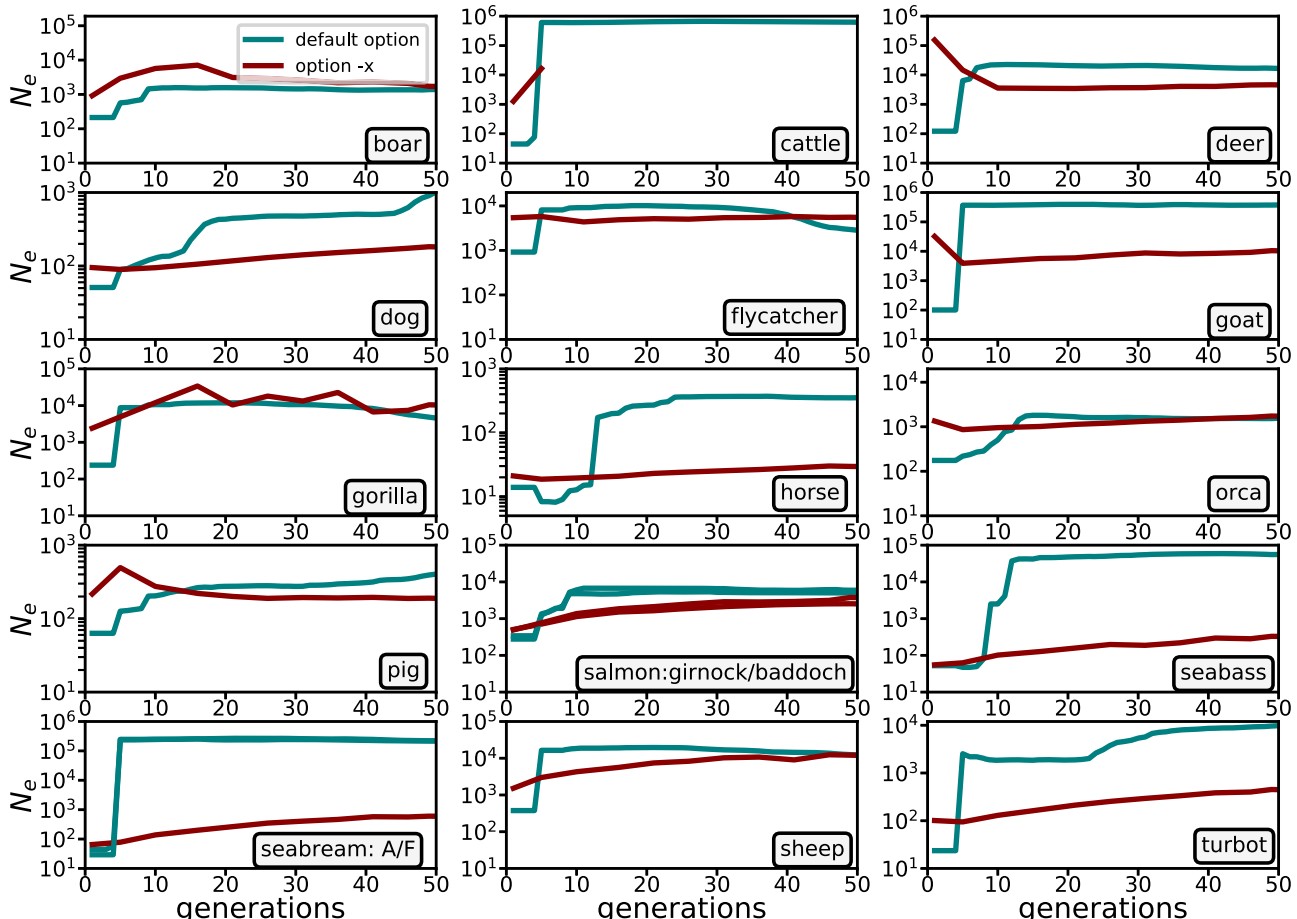

**Fig. 3 | Historical $N_e$ estimates for different species using *GONE2*.** Estimates of $N_e$ trajectories over the last 50 generations for the same samples analyzed in Table 1. Each panel corresponds to a different species, with estimates obtained using *GONE2* under two assumptions: the default option (green lines), which assumes panmixia, and the '-x' option (red lines), which takes population structure into account. The salmon panel combines the analyses of two tributaries of the river Dee. Seabream panel combines the analysis from two parallel breeding programmes. *GONE2* did not produce results for cattle for more than five generations, probably due to the complexity of the mixing of 134 highly differentiated breeds. Source data are provided as a Source Data file.

panmixia and assuming structure are close. Single-breed or experimental population samples show similar $N_e$ estimates between methods, with minor differences for horse and turbot. There is a fairly clear linear relationship between the ratio of the two estimates and the migration rate estimate (Supplementary Fig. 4). This is somewhat to be expected as both estimates depend on the inverse of the LD. In differentiated subpopulations, assuming panmixia, the LD between unlinked loci is dominated by the square of the $F_{ST}$, which is quite proportional to the migration rate. However, this dependence on the $F_{ST}$ is eliminated from the $N_e$ estimate when structure is assumed. This difference leads to a proportionality of the ratio to the migration rate.

Contemporary $N_e$ estimates using *currentNe2* can be combined with historical $N_e$ estimates using *GONE2* (Fig. 3) to infer the recent past demography. The demographic histories of the two salmon populations and the domestic populations of seabass and seabream, which are found to be essentially panmictic, are better represented by the *GONE2* predictions assuming panmixia (green lines), while the prediction assuming structure (option '-x') shows only the general demographic trend. For the other species, the trend resulting from the structure assumption analysis is the best option, especially for those with large differences in estimates from Table 1.

*GONE2* now addresses biases in $N_e$ estimates from low quality DNA. Direct analysis of low-depth sequencing or error-prone genotyping data inflates historical $N_e$ estimates, but appropriate corrections effectively mitigate these biases (Fig. 2, rows 3 and 4). The genotyping

error correction requires known error rates, which can be estimated in some contexts[28,36], whereas the sequencing depth correction is adaptable to any depth.

Figure 4 shows $N_e$ estimates from autosomes and from X-chromosome sequencing data of *Drosophila* males from four laboratory populations[37] using the new haploid analysis option of *GONE2*. These estimates (at the diploid scale) tend to be higher than those from autosome data of the same males, especially in the most recent generations, for which $N_e$ is generally estimated at higher resolution. This suggests that despite the equal number of sexes imposed by management, the effective number of reproducing males was lower than the number of females, leading to an increased representation of X chromosomes and higher diploid census estimates after correction (see Methods).

## Discussion

*GONE2* and *currentNe2* now account for population subdivision in their estimates of effective population size, addressing a fundamental challenge in population genetics. Natural populations often exhibit some degree of structure that can significantly bias $N_e$ estimates if not properly accounted for. The inclusion of genetically distinct individuals from multiple demes in a single sample amplifies observed linkage disequilibrium[38], analogous to a two-locus version of the Wahlund effect[39]. Neglecting population structure introduces an artefact in temporal $N_e$ estimates, characterized by an apparent sharp

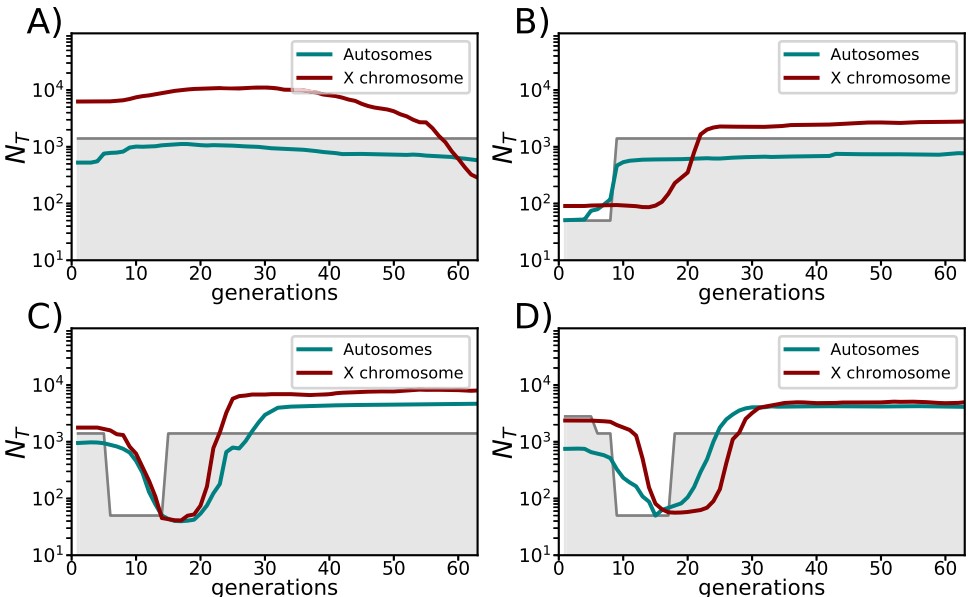

**Fig. 4 | Estimates of the historical $N_e$ using *GONE2* with samples from 17 males of *Drosophila melanogaster*.** The experimental populations were maintained at constant size (**A**), underwent a recent drastic reduction in size (**B**), recovered from a previous bottleneck (**C**), and underwent further expansion after a bottleneck (**D**), as shown by the grey shaded areas. The green lines show the estimates of $N_e$ from autosomes using the modifier '-g 0' for unphased diploid genotypes. The red lines show the estimates of $N_e$ from X chromosomes using the haploid option '-g 1'. The $N_e$ estimates from X chromosomes have been transformed to the autosomal scale (see Methods) to compare sex ratios. Source data are provided as a Source Data file.

decline in the generations closest to the sampling time[18,24,40], which also affects contemporary $N_e$ estimation tools[20,21]. If the '-x' option yields larger $N_e$ estimates for current generations than estimates with the default option, it serves as an indicator of population subdivision. This has implications for conservation genetics, as populations previously deemed at risk of extinction ($N_e < 500$)[26] may in fact have larger $N_e$ values if population subdivision is properly accounted for in the estimation.

While Ragsdale and Gravel[27] have comprehensively described the LD evolution in multi-population systems, the development of a computationally efficient method to estimate recent historical $N_e$ of metapopulations from a single sample of individuals has remained elusive. Our approach addresses this problem by combining three observations from a single sample: LD of unlinked loci, LD of weakly linked loci and the average inbreeding coefficient. Under the island model, we estimate five parameters: the effective size of the metapopulation ($N_e$), the sum of the effective sizes of the subpopulations ($N_T$), the genetic differentiation index ($F_{ST}$), the migration rate ($m$), and the number of subpopulations ($s$). These parameters are reduced to three unknowns by two key relationships: Wright's[32] ratio $N_T/N_e = (1 - F_{ST})$ and Takahata's[33] $F_{ST}$ equation, the latter linking all parameters except $N_e$ at migration-drift equilibrium. This system of equations yields a computationally tractable solution with remarkable robustness. It provides accurate contemporary estimates of $N_e$ even when key assumptions are violated, such as deviations from the island model, unequal sampling, or heterogeneous subpopulation sizes. Migration rate estimates are also accurate when subpopulation differentiation is high. However, when $Nm \gg 1$ (where $N$ is the subpopulation size), migration rates tend to be underestimated, probably due to the inherent difficulty in achieving precision as $F_{ST}$ values approach zero. Notably, the method shows accuracy in estimating subpopulation numbers, except in stepping stone model scenarios.

Our research reveals a compelling relationship: the linkage disequilibrium ($\delta^2$) between loci on different chromosomes ($c = 0.5$) closely approximates the ratio between the square of the $F_{ST}$ and the number of subpopulations. This finding lies in the core of the method and provides a way of estimating the migration rate from a single sample of individuals. The approach complements existing techniques, such as those proposed by Waples[41] and Zhivotovsky[42], which primarily use deviations from the Hardy-Weinberg equilibrium within a sample.

If the migration rate is large (e.g. $Nm > 10$), sampling from a single subpopulation will approximate the estimate for the whole metapopulation. However, analyzing samples from a single subpopulation within a highly differentiated metapopulation is very similar to analyzing a panmictic population with the size of the subpopulation, as noted by Waples and England[22]. In this case, the default analysis option in *GONE2* or *currentNe2* demonstrates remarkably robustness in estimating the effective size of the subpopulation, a finding further supported by Novo et al.[24]. Notably, the '-x' option, designed to account for metapopulation structure, does not improve resolution in such scenarios. Instead, it may introduce unnecessary complexity, potentially obscuring the true demographic patterns of the sampled subpopulation. This observation underscores the importance of understanding the sampling context when choosing the method of analysis, and highlights that the more complex metapopulation model is not always advantageous, particularly when dealing with samples from a single, relatively isolated subpopulation within a broader metapopulation framework. In this context, comparing confidence intervals of estimates using *currentNe2* could be helpful in determining the most appropriate model when the population structure is unknown or the origin of the samples is unclear. Applying this approach to the set of species in this study suggests a general interpretation: natural populations typically exhibit strong population structures (i.e. $Nm \ll 10$). The salmon populations from the river Dee may be an exception, although their classification as truly natural populations is debatable due to restoration management practices. For domestic species, analyses of samples containing a mixture of breeds consistently show clear separation of confidence intervals using *currentNe2*. Conversely, samples from pure breeds or single managed populations tend to show overlapping or nearly overlapping intervals.

*GONE2* now addresses two key challenges related with low-quality DNA genotyping: low sequencing coverage and base calling errors. These issues can artificially inflate $N_e$ estimates due to allele

detection failures. The software incorporates robust corrections that, when properly applied, effectively mitigate these biases. The sequencing depth solution is particularly versatile, adapting to various depth distributions without parametric constraints, ensuring accurate $N_e$ estimation across diverse data quality scenarios. Furthermore, *GONE2*'s new capability to analyze haploid data significantly broadens its applicability in genetic studies. This feature is particularly valuable for research on haploid species, organisms with haplodiploid sex determination systems, and for investigating sex ratios in diploid species.

By accounting for population structure and genotyping error, this framework extends the operational scope of demographic inference in both natural and managed populations. This is particularly relevant for non-model species, since a precise genetic map is not required for the analysis. Future integrations with environmental covariates could further enhance the ecological relevance of $N_e$ estimation, contributing to its transformation from a population genetic abstraction into a practical conservation metric.

## Methods

### Working with the software
*GONE2* and *currentNe2* are C++ programs designed for Linux environments, using a command line interface. These tools use a user-friendly syntax: 'gone2 <modifiers> filename' and 'currentne2 <modifiers> filename', where *filename* represents the input sample data file. The programs support the VCF format[43], and the PED or TPED formats[44]. *GONE2* processes genotyping data to compute observed $d^2$ values in bins of recombination rate *c* between SNP pairs. Since PED files do not contain genetic map information, locations must be available in an accompanying map file in PLINK format. A genetic map can be approximated by a physical map if a whole-genome recombination rate is provided to the program (option '-r'). *CurrentNe2* uses the general Eq. (1) in Santiago et al. [21], which takes into account the mating structure. If no information on marker locations are available in the input files, it performs a single estimate of contemporary $N_e$ assuming that markers are evenly distributed across the genome. If marker assignments to chromosomes are available, it additionally provides estimates using the observed $\delta^2$ values between sites on different chromosomes. In addition, if the genetic locations of the markers are available, *currentNe2* will use them in the estimates, otherwise, it would convert physical locations to genetic locations by assuming a constant recombination rate across the genome (option '-r'). Both programs offer the '-x' modifier for metapopulation structure analysis, which is based on Eq. (1) as explained before. The algorithm shown in Supplementary Fig. 5 is used to estimate the three relevant parameters *N* (the subpopulation size), *s* and *m*. $F_{ST}$ is calculated from these values using the equation of Takahata[33]. The total population size and the effective population size of the metapopulation are calculated as $N_T = sN$ and $N_e = N_T/(1 - F_{ST})$ respectively. *GONE2* introduces additional features to handle different scenarios: base calling errors with the '-b error_rate' modifier, low sequencing depth with the '-g 3' modifier and haploid populations with the '-g 1' modifier. Further details of the software implementations can be found in their respective online repositories.

*GONE2* does not generate confidence intervals for $N_e$ estimates. However, for datasets with a large number of chromosomes, empirical confidence limits can be obtained using subsets of chromosomes. These intervals should be interpreted with caution, as they do not account for pedigree sampling effects in small samples. Alternatively, when the sample size is sufficiently large, confidence intervals can be obtained empirically by generating replicate estimates from individual subsamples. In contrast, *currentNe2* generates confidence intervals for $N_e$ estimates under panmixia using the approach of Santiago et al.[21], which is also applied to subdivided populations for guidance.

### Computer simulations
To test the new features of the software we used the SLiM v3 software[45]. The genomes generally consisted of 20 chromosomes, each 100 Mb and 100 cM long, with neutral mutations occurring at a rate between 1.4 and $17 \times 10^{-9}$, resulting in ~50,000 SNPs available for analysis in each sample. We ran simulations for 10,000 discrete generations with different population sizes depending on the scenario. At the end of the simulation, 100 individuals were sampled to estimate $N_e$. In some of the scenarios, base calling errors were introduced at a rate of 5% by randomly replacing one allele with the other. To simulate low coverage, we converted heterozygotes to homozygotes at a rate of 1/2, which is roughly equivalent to an average read depth of 2× after filtering out sites with no reads.

### Empirical analyses
*GONE2* was tested with sequencing data from four samples, each consisting of 17 males, from four laboratory populations of *Drosophila melanogaster* provided by Novo et al.[37]. These populations were maintained with controlled numbers of adults (40–50 individuals of each sex per vial), either constant or with recent significant changes in size. We estimated historical haploid $N_e$ using the haploid option '-g 1 ' of *GONE2* with X chromosome data from these males. The genetic map, which is based on female meiosis, was corrected by reducing the recombination rates by 2/3 (males only carry one copy, so 1/3 of chromosomes do not recombine). This was done to calculate the effective genetic distances for X chromosomes. The recombination rates of autosomes were corrected by multiplying by 1/2 because meiosis of *Drosophila* males is achiasmatic (males carry two copies, so 2/4 of the chromosomes do not recombine). For comparison with autosomal results from the same males in order to get sex ratios, these haploid $N_e$ estimates from X chromosomes were rescaled by multiplying by 4/3, to obtain the haploid equivalent in autosomes with equal sex numbers, and dividing by 2 to restore the diploid number. The number of SNPs available for analysis ranged from 85,767 to 143,742 across the four samples.

We used *currentNe2* and *GONE2* to estimate contemporary and historical $N_e$ for different animal species, comparing results assuming geographic structure (option '-x') with those assuming panmixia (default option). Genotyping data were obtained from supplementary material of the original publications (Table 1), typically as VCF or PED files containing SNP genotypes and locations. However, for gorillas, orcas, and deer only *fastq* sequencing data were available, which were processed using *Trimmomatic*[46] for trimming, and *minimap2*[47] for mapping to reference genomes (gorilla: assembly GCF_029281585.2; deer: assembly GCA_014726795.1; orca: assembly CNA0050865). The *SAMtools* and *BCFtools* utilities[48] were used to filter unmapped reads and duplicates (flags '-F 1804'), calculate genotype likelihoods (quality options '-Q 20 -q 20'), perform variant calling (multiallelic) and additional filtering (-e 'DP < 20||DP > 2*MEAN(DP) || AD[GT] <= 5'), excluding SNPs with missing genotypes. Chromosomes or scaffolds smaller than 20 Mb were discarded. In the absence of species-specific genetic maps, a uniform recombination rate of 1 cM/Mb (option '-r 1') was applied, except for turbot (2.47 cM/Mb), seabream (2.13 cM/Mb), seabass (2.38 cM/Mb) [49], and gorilla (1.19 cM/Mb)[50].

### Reporting summary
Further information on research design is available in the Nature Portfolio Reporting Summary linked to this article.

## Data availability
The individual data points corresponding to the figures in the main text, along with links to all relevant datasets and accession codes used in this study, are detailed in the supplementary file Source_Data.xlsx. Scripts for SLiM simulations can be accessed on the GitHub address

https://github.com/esrud/currentNe2/blob/main/slim_simulation_inputs. Source data are provided with this paper.

## Code availability

The code of the software programmes *GONE2* and *currentNe2* developed and used to perform the analyses and generate results in this study are available at GitHub (https://github.com/esrud/GONE2[51] and https://github.com/esrud/currentNe2[52]), under a GPL-3.0 license. Users are permitted to reuse, modify, and distribute the code in accordance with the terms of the license.

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

## Acknowledgements

This study forms part of the Marine Science Programme (ThinkInAzul) supported by the Ministerio de Ciencia e Innovación and Xunta de Galicia with funding from the European Union NextGenerationEU (PRTR-C17.I1) and European Maritime and Fisheries Fund (A.C.). We also acknowledge financial support from grants PID2020-114426GB-C21 (MCIN/AEI/10.13039/501100011033) (A.C., E.S.), Xunta de Galicia (ED431C 2024/22) (A.C.), Centro Singular de Investigación de Galicia accreditation 2024-2027 (ED431G 2023/07) and "ERDF A way of making Europe" (A.C.). We thank Mary Riádigos for administrative support.

## Author contributions

E.S. and A.C. conceived the work and wrote the manuscript. E.S. developed the theory. E.S. and C.K. developed the software. A.C. conducted simulations and analyses. C.K. led the bioinformatics processing. All authors revised the manuscript.

## Competing interests

The authors declare no competing interests.
