## [Transparent Peer Review file · Nature Communications]

Accounting for Population Structure and Data Quality in Demographic Inference with Linkage Disequilibrium Methods

Corresponding Author: Dr Enrique Santiago

Version 0:

Reviewer comments:

Reviewer #1

(Remarks to the Author)

Santiago et al present updated methods to infer effective population sizes using linkage disequilibrium. The key innovation of this new work is that they develop approaches that are robust to meta-population structure. This is done by deriving the equations for equilibrium LD under a model of two populations, and then using those in either a moment-based approach currentNe2 or a HMM approach with GONE2. Interestingly, the authors show using simulations that while the methods are developed with two demes in mind, they seem to apply to island models with more demes, and with other violations of the assumptions (such as unequal population or sample sizes). They also show that there is a pretty substantial difference when applied to real data in many cases, with effective sizes being estimated to be much larger when accounting for meta population structure. They also have some additional features to account for sequencing error or low depth of coverage.

This is an interesting and useful contribution to the literature, as certainly many natural populations exist with some form of population structure. The authors simulations show that meta population structure results in under-estimation of effective sizes when there is population structure, in line with the results in real data. The authors suggest that a difference between the population size estimated with and without the assumption of structure can actually be used as a test for population structure. I think that it would enhance the paper to more fully look at the properties of that test: for example, if you reject panmixia when the CIs for Ne between the two methods don't overlap, what are the size and power of the test? I believe the authors can do this using the simulated data they already have, so I do not anticipate it being a major burden, but it will strengthen the paper.

I think another small and hopefully easily fixable issue with the manuscript is that though the workings of currentNe2 are described in some detail (although I have a couple clarifying questions), I don't believe the workings of GONE2 are described in much detail in the manuscript. I think some description of the algorithm, along the lines of the description of the currentNe2 algorithm, would be helpful, even if it's just in the supplement. There's some description for the GONE2 algorithm in the discussion, which feels like an odd place for it.

Related to both those questions, it doesn't seem that GONE2 has a method of determining whether using the panmictic option or the meta population option is appropriate, as is the case for currentNe2. I feel that some discussion of that point would be appropriate.

I have some more minor comments as well

1) The software described uses only sites on different chromosomes and sites at $c = 0.05$ when they're on the same chromosome. This seems to be throwing out a lot of data---for example, could currentNe2 use a bunch of different c and a regression/least squares approach to get more out of the data?

2) GONE2 with errors has to have an error rate pre-specified. In some other contexts (e.g. Racimo et al 2016, Schraiber 2018), it's possible to estimate the error rate. While those are very different models that probably gain identifiability by being "anchored" to "true allele frequencies" in some population, I wonder if there's any hope of estimating an error rate in the GONE2 model? If not, can the authors provide some guidance on how to set the error rate?

3) I wonder if, for the real data analysis presented in Table 1, a scatterplot of m vs the log ratio of Ne's estimated from

panmixia vs meta population (i.e. $\log(Ne_{\text{panmixia}}/Ne_{\text{metapop}})$) could be an interesting visualization of that table, to help people understand what's going on.

4) Do the dashed horizontal lines in Table 1 represent some important breaks in the datasets? I couldn't really tell

5) Fig 1: why are simulations summarized as geometric mean instead of arithmetic mean?

I prefer to sign my reviews. My name is Joshua Schraiber

References:

Racimo, F., Renaud, G., & Slatkin, M. (2016). Joint estimation of contamination, error and demography for nuclear DNA from ancient humans. *PLoS genetics*, 12(4), e1005972.

Schraiber, J. G. (2018). Assessing the relationship of ancient and modern populations. *Genetics*, 208(1), 383-398.

(Remarks on code availability)

Reviewer #2

(Remarks to the Author)

The authors present new theory and software updates for two previously introduced tools, GONE and currentNe. These tools use LD patterns to estimate very recent effective population sizes. In their updates, both software tools now account for substructures populations and estimate Ne and a migration rate. The paper conducts simulations and applications to real data from *Drosophila* to test the method.

These are interesting developments on previously successful tools, both in terms of theory and implementation. However, I found the paper quite hard to read, assuming in many places detailed knowledge about how their previous versions worked. It is unclear to me how general the proposed model is, and therefore how applicable it is to real scenarios: for instance how does this work if there are multiple subpopulations of different size? As a user, I would like to see more extensive testing on real applications, currently there is only one application of GONE2 and none for currentNe. It would be nice to see the difference in inference between the previous versions and these updates, and comparison with existing recent software tools such as HapLD. More details on e.g. the number of genomes needed for both tools would be good too.

L. 35: "metapopulation from a single sample": In my first parse through the paper, I got really confused because this sounded like you can apply the method to a single genome. That seemed strange given you rely on LD. I am still unsure what exactly you mean by sample, but I would definitely consider changing the phrasing here.

L. 70: "Infer current Ne from a relatively small number of unphased genotypes": How many? And compared to what?

L. 98: Here I would add at least a section on how GONE and currentNe work. I had to go back to your other papers.

L. 109: I am confused by the statement: "we show that a two-population model with reciprocal migration rate m can effectively approximate the effective size of the entire metapopulation using a single sample": Is the claim that even if you have more than 2 populations in reality, your approach infers Ne and migration rates correctly assuming only 2 populations? That seems quite strange to me, and would also presumably depend on how different migration rates are between different populations, differences in their Ne etc?

L. 147: This is very brief and mostly refers to a previous paper and I couldn't follow what exactly the inference steps are. Does this only apply to GONE or to both methods?

L. 160: It would be nice to get an overview of how this pseudohaploid version works, here in the main text. I assume you have to do a correction for potentially sampling the other haploid sequence at different sites? Does this also work for currentNE?

L. 184: Comparing how? How do we know if something is 'significantly different'?

Figure 2:

- Why does the -x option perform badly under panmixia, for variable Ne? Shouldn't -x be nested within the default option parameter space? In practice, I will apply both options, notice a difference and then go with -x as suggested which is less accurate.

Figure 3:

- I'd appreciate a bit more discussion on Figure 3 in the main text. In my understanding males don't recombine, is that correct? That seems like will have an effect on LD? Can this not explain the patterns you see, of X chromosomes having higher Ne?

(Remarks on code availability)

Code looked good to me! I tried to install it, ran into errors on my Mac.

I changed the following to the makefile to get it working:

```
COMMON_FLAGS=-Wall -Xpreprocessor -fopenmp -lomp -l/opt/homebrew/Cellar/libomp/19.1.5/include -  
L/opt/homebrew/Cellar/libomp/19.1.5/lib -std=c++11
```

Reviewer #3

(Remarks to the Author)

The authors of this MS describes the extensions made to their previously proposed methodologies for estimating the current and historical effective population sizes N_e from linkage disequilibrium (LD) in marker data. The extensions are made so that the methods apply to subdivided populations for the estimation of both N_e and migration rate (m), and to low quality marker data that either are of low sequencing depth or have genotyping errors. The accuracy and the robustness of these extended methods are checked by analysing some simulated data, and are also demonstrated by applying to the analysis of drosophila experimental populations with known demography. I believe this research represents significant developments in estimating population demography based on LD in genomic marker data.

I have just one major and a few minor comments for the authors to consider in revision of this MS.

The major development of this research is to extend the previous LD-based N_e estimation method to apply to subdivided populations. In doing so, it is assumed that a population is subdivided into 2 equal-sized (identical) subpopulations with symmetrical migration rates m (the same m from subpopulation 1 to 2, and from 2 to 1). This simplest subdivision model simplifies the parameter estimation tremendously. However, real populations might be subdivided in many different ways, with a variable number of subpopulations and variable migration rates among subpopulations. These subdivision details do matter with regard to the genetic structure of the population and thus to the N_e of the subdivided population. For example, the number of subpopulations, n , partly determines the N_e . If a population is subdivided into n subpopulations in Wright's (1943) island model, each subpopulation being an idealized population of size N except for receiving a proportion m of immigrants taken randomly from the entire population per generation, the N_e of the population is $N_e = nN(1 + (n-1)^2 / (4Nm))$ derived by Nei & Takahata (1993). Under this model, for a given total population size nN , the higher is the number of subdivisions n , the larger will be the value of N_e . Therefore, assuming $n=2$ in the LD-based method might lead to a biased estimate of N_e .

L126, the parameter m in the equation is first encountered herein without explanation. What does it represent? Migration rate?

L133, the symbol ΔT^2 is not explained.

L179, "analysis option ('-x') is used", where "-x" is not explained. What does it mean?

L279-283, analysing a sample containing individuals sampled from a single subpopulation leads to an estimate of the N_e of the sampled subpopulation, not the entire population. Is this always true, regardless of the value of m ? I think if m is sufficiently large that the subdivided population behaves like a panmictic population, then the LD analysis of a sample of individuals from a single subpopulation should still lead to an estimate of N_e of the entire population.

In Figure 1 and other places, the geometric mean estimates of N_e was used and compared with the simulated value. Why not using arithmetic mean which is commonly used to show the biasness of a method?

In Figure 1B, a population of 2000 individuals is subdivided into n equally sized subpopulations with a migration rate $m = 0.001$. The simulated (theoretical) N_e of the population is about 2000, invariable with the value of n (2-7) as shown in 1B. However, under this subdivision model, the simulated (theoretical) N_e values calculated from Nei & Takahata (1993) formula are 2125, 2333, 2562, 2800, 3041 and 3285 for $n=2,3,4,5,6,7$ respectively.

In Figure 1D, "Estimates for a metapopulation composed of two unequally sized subpopulations (total 2,000 individuals, $m = 0.001$)." The simulated (theoretical) N_e of the population is about 2000, invariable with the extent of the unbalance in subpopulation size. Is this true?

(Remarks on code availability)

I did not try the code, and did not review the derivations.

Version 1:

Reviewer comments:

Reviewer #1

(Remarks to the Author)

I am very happy with the authors' responses to my comments as well as the other reviewers. I really think the authors went

above and beyond by modifying the model to explicitly model the number of populations and estimate that number. That's really cool. I think the organization of the manuscript is also greatly improved. I think this is a very nice manuscript.

(Remarks on code availability)

Reviewer #2

(Remarks to the Author)

The revised manuscript addresses all of my comments and I think the advances presented for the software tools GONE and currentNe are significant and not addressed by alternative methods. Thank you for your responses. I found the manuscript now much clearer to parse and it describes the methods in more detail than before.

- The manuscript extensively tests the proposed methods using simulations and I find the additional simulations presented with several subpopulations and including varying size (SI Fig 2, Table S4) are very nice. Small comment, in Figure 1 it is a bit hard to spot initially that currentNe1 is off the scale. Table S4 actually illustrates the difference more clearly.

- I very much appreciate the addition of the section "Working with the software" and the added explanation of how the two methods work in the main text.

- I'm sorry I missed Table 1 for some reason previously, this is quite remarkable.

- Code: Please feel free to add the makefile modification to the repo.

(Remarks on code availability)

Reviewer #3

(Remarks to the Author)

The authors have revised their MS by addressing my comments and have improved the work substantially. For example, now the new method provides estimates of Ne (in addition to Nt) of a metapopulation and the number of subpopulations in a metapopulation. The presentation has been proved as well. In my opinion, the MS is acceptable for publication.

(Remarks on code availability)

It is straightforward to install the code on my laptop (WSL). A test run indicates that it works as expected.

RESPONSES TO REVIEWER COMMENTS:

NOTE: The original title (“Advanced Linkage Disequilibrium Methods for Demographic Inference: From Metapopulations to Poor Quality DNA Data”) has been changed to “Accounting for Population Structure and Data Quality in Demographic Inference with Linkage Disequilibrium Methods” to comply with the journal's formatting guidelines.

Reviewer #1 (Remarks to the Author):

Santiago et al present updated methods to infer effective population sizes using linkage disequilibrium. The key innovation of this new work is that they develop approaches that are robust to meta-population structure. This is done by deriving the equations for equilibrium LD under a model of two populations, and then using those in either a moment-based approach currentNe2 or a HMM approach with GONE2. Interestingly, the authors show using simulations that while the methods are developed with two demes in mind, they seem to apply to island models with more demes, and with other violations of the assumptions (such as unequal population or sample sizes). They also show that there is a pretty substantial difference when applied to real data in many cases, with effective sizes being estimated to be much larger when accounting for meta population structure. They also have some additional features to account for sequencing error or low depth of coverage.

RESPONSE:

We thank the reviewer for all his comments. They have helped us to rethink several parts of the theory, the programmes and the manuscript.

This is an interesting and useful contribution to the literature, as certainly many natural populations exist with some form of population structure. The authors simulations show that meta population structure results in under-estimation of effective sizes when there is population structure, in line with the results in real data. The authors suggest that a difference between the population size estimated with and without the assumption of structure can actually be used as a test for population structure. I think that it would enhance the paper to more fully look at the properties of that test: for example, if you reject panmixia when the CIs for Ne between the two methods don't overlap, what are the size and power of the test? I believe the authors can do this using the simulated data they already have, so I do not anticipate it being a major burden, but it will strengthen the paper.

RESPONSE:

Thank you for your insightful comment. We have addressed the suggestion to explore the properties of the test for detecting population structure by comparing the confidence intervals of Ne estimates with and without assuming structure. The additional simulations, detailed in Supplementary Table S4, used an island model with five subpopulations, each with N=400 or N=2000 individuals, and varying migration rates (m). The results show that the method can reject panmixia for $Nm < 10-20$, which is a substantial number of migrants

per generation, indicating that our approach is sensitive to detecting population structure even under moderate to high gene flow.

I think another small and hopefully easily fixable issue with the manuscript is that though the workings of currentNe2 are described in some detail (although I have a couple clarifying questions), I don't believe the workings of GONE2 are described in much detail in the manuscript. I think some description of the algorithm, along the lines of the description of the currentNe2 algorithm, would be helpful, even if it's just in the supplement. There's some description for the GONE2 algorithm in the discussion, which feels like an odd place for it.

RESPONSE:

(this response is common for reviewers 1 and 2)

We have addressed this by providing additional information on how both currentNe2 and GONE2 work to estimate effective population sizes when dealing with metapopulations:

1- We have extended the explanation of the algorithm used to solve the current metapopulation structure with both programs. This includes modifications to Figure S1, additional explanations between lines 119 and 142, and an extension of the "Working with the Software" section in Methods (lines 370-376), which now references a new Figure S5. This figure provides the pseudocode of the algorithm used to determine the unknowns describing a metapopulation..

2- The description of the GONE2 algorithm for estimating temporal effective population sizes in metapopulations has been relocated to lines 143-159, immediately following the introduction of the algorithm for solving current metapopulation structure. This change connects the descriptions of both algorithms, providing a clearer explanation.

Related to both those questions, it doesn't seem that GONE2 has a method of determining whether using the panmictic option or the meta population option is appropriate, as is the case for currentNe2. I feel that some discussion of that point would be appropriate.

RESPONSE:

Unlike currentNe2, GONE2 does not provide a method to decide between models based on confidence intervals. Instead, GONE2 relies on subjective interpretation of the results, such as the proximity of the Ne estimates in the first generation and the presence of a typical drop in historical Ne around generation 5 when using the default option. Both programmes are sensitive to detecting population structure, and combining them is a good option as they are complementary.

I have some more minor comments as well

1) The software described uses only sites on different chromosomes and sites at $c = 0.05$ when they're on the same chromosome. This seems to be throwing out a lot of

data---for example, could currentNe2 use a bunch of different c and a regression/least squares approach to get more out of the data?

RESPONSE:

When the aim is to estimate the current N_e , we require at least two measures of LD that reflect similar drift events, and thus correspond to the same N_e . Most LD between chromosomes ($c=0.5$) or between weakly linked sites ($c>0.05$) reflects very recent drift (L129-139), allowing us to assume that they share the same N_e . While the estimate of LD between chromosomes is a simple average, the LD within chromosome is obtained by integration over all the loci pairs with rec. rates $c>0.05$, giving both points equal weight. We explored alternative approaches before, but the inherent large errors of the measures of LD make them complex and difficult to integrate into the parameter estimation process.

2) GONE2 with errors has to have an error rate pre-specified. In some other contexts (e.g. Racimo et al 2016, Schraiber 2018), it's possible to estimate the error rate. While those are very different models that probably gain identifiability by being "anchored" to "true allele frequencies" in some population, I wonder if there's any hope of estimating an error rate in the GONE2 model? If not, can the authors provide some guidance on how to set the error rate?

RESPONSE:

We believe this is somewhat out of the scope of our software. However, we now cite these methods (L262) to estimate the error rate, which then could be specified when running software.

3) I wonder if, for the real data analysis presented in Table 1, a scatterplot of m vs the log ratio of N_e 's estimated from panmixia vs meta population (i.e. $\log(N_e_{\text{panmixia}}/N_e_{\text{metapop}})$) could be an interesting visualization of that table, to help people understand what's going on.

RESPONSE:

We have created this plot now included as supplementary Figure S4. It shows a linear relationship. The argument explaining this relationship is now included in the manuscript (L243-249).

4) Do the dashed horizontal lines in Table 1 represent some important breaks in the datasets? I couldn't really tell

RESPONSE:

The lines have been removed (they were included only to read more easily the figures across lines).

5) Fig 1: why are simulations summarized as geometric mean instead of arithmetic mean?

RESPONSE:

(this response is common for reviewers 1 and 3)

We chose the geometric mean because the sampling error of linkage disequilibrium (LD) is approximately normally distributed on a log scale. Also, LD is actually the inverse of N_e , and the right way to transform this relationship into a linear function is using a log scale. Transforming to logs, calculating the arithmetic mean and applying antilogs, is the same as calculating the geometric mean. The geometric mean offers advantages over the arithmetic mean by being less sensitive to outliers and providing a more robust measure of central tendency in skewed distributions.

Reviewer #2 (Remarks to the Author):

The authors present new theory and software updates for two previously introduced tools, GONE and currentNe. These tools use LD patterns to estimate very recent effective population sizes. In their updates, both software tools now account for substructures populations and estimate N_e and a migration rate. The paper conducts simulations and applications to real data from *Drosophila* to test the method.

RESPONSE:

We thank the reviewer for all the comments. They have helped us to improve the tools and the manuscript.

These are interesting developments on previously successful tools, both in terms of theory and implementation. However, I found the paper quite hard to read, assuming in many places detailed knowledge about how their previous versions worked.

RESPONSE:

We have rewritten and reorganised the explanations of how both tools work and their connections. You can find these in lines 119 to 159. We have also included Figure S5, which shows the pseudocode of the algorithm used to estimate effective population size in metapopulations. We hope these changes make the paper easier to read.

It is unclear to me how general the proposed model is, and therefore how applicable it is to real scenarios: for instance how does this work if there are multiple subpopulations of different size?

RESPONSE:

Thank you for this comment, which has encouraged us to extend the model so that it can now consider any number of subpopulations. This has now been added to both programs so that an effective number of subpopulations can be estimated. We have also done some new simulations to try out your suggestions. These include an island model with different subpopulation sizes, a stepping stone model with different numbers of subpopulations, an island-continent migration model, and a model with asymmetric migration between two islands. You can see these simulations in Supplementary Figure S2.

As a user, I would like to see more extensive testing on real applications, currently there is only one application of GONE2 and none for currentNe.

RESPONSE:

We give a substantial number of applications of currentNe2 to real data in our Table 1. The table has been improved and updated by using the latest version of the software, and now provides the estimated number of subpopulations (s) and an estimate of the migration rate and the fixation index (F_{st}). In addition, we have now added estimates of historical N_e with GONE2 in Figure 3 for the examples of Table 1. These results are now commented in lines 250 to 257

It would be nice to see the difference in inference between the previous versions and these updates, and comparison with existing recent software tools such as HapLD.

RESPONSE:

The current version of GONE2 gives the same (or very similar) results as the previous version (GONE) if there is not population structure. But the previous version does not account for genotyping errors and low sequencing depth. Likewise, currentNe2 gives the same results as the previous version (currentNe) if there is not population structure. This is now mentioned in the paper to avoid confusions (lines 205 and 217

Regarding HapNe-LD, which is tailored for human data, we have included a comparison with GONE2 in Supplementary Figure S3 and a comment in line 231. This comparison involves simulations mimicking human chromosomes, highlighting the broader applicability of our tools across different species. We appreciate your feedback and hope this addition enhances the paper's interest.

More details on e.g. the number of genomes needed for both tools would be good too.

RESPONSE:

It is a difficult to give a number for a reliable result, which depends on the number of individual and markers in the sample, the population size and the genome size. This problem was addressed in Santiago et al. (2024), resulting in the CIs implemented in currentNe and currentNe2.

L. 35: “metapopulation from a single sample”: In my first parse through the paper, I got really confused because this sounded like you can apply the method to a single genome. That seemed strange given you rely on LD. I am still unsure what exactly you mean by sample, but I would definitely consider changing the phrasing here.

RESPONSE:

Thank you for pointing out the potential confusion. Now we make clear that when we say 'single sample' we mean a set of individuals every time we use it.

L. 70: "Infer current N_e from a relatively small number of unphased genotypes": How many? And compared to what?

RESPONSE:

The sentence is imprecise and a bit unfortunate. As indicated in response to a previous comment, there are several factors affecting accuracy (sample sizes of individuals and polymorphic markers, the true population size and the genome size), which make it impossible to give a number valid in all cases. We have decided to remove the sentence. We think that the range of sample sizes in Table 1 gives an indication of the applicability of the method.

L. 98: Here I would add at least a section on how GONE and current N_e work. I had to go back to your other papers.

RESPONSE:

(this response is common for reviewers 1 and 2)

We have addressed this by providing additional information on how both current N_e 2 and GONE2 work to estimate effective population sizes when dealing with metapopulations:

1- We have expanded the explanation of the algorithm used to solve the current metapopulation structure with both programs. This includes modifications to Figure S1, additional explanations between lines 119 and 142, and an extension of the "Working with the Software" section in Methods (lines 370-375), which now references a new Figure S5. This figure provides the pseudocode of the algorithm used to determine the unknowns describing a metapopulation..

2- The description of the GONE2 algorithm for estimating temporal effective population sizes in metapopulations has been relocated to lines 143-159, immediately following the introduction of the algorithm for solving current metapopulation structure. This change connects the descriptions of both algorithms, providing a clearer explanation.

L. 109: I am confused by the statement: "we show that a two-population model with reciprocal migration rate m can effectively approximate the effective size of the entire metapopulation using a single sample": Is the claim that even if you have more than 2 populations in reality, your approach infers N_e and migration rates correctly assuming only 2 populations? That seems quite strange to me, and would also presumably depend on how different migration rates are between different populations, differences in their N_e etc?

RESPONSE:

Assuming two subpopulations gave quite good estimates of N_e when the true subpopulation number was larger, although estimates progressively deviated to underestimation as the number increases. The good prediction is partly due to fact that the model was built to predict N_T , regardless how individuals are distributed among subpopulations. However, the F_{st} estimates were less

accurate. The updated model now considers any number of subpopulations, leading to slightly overestimated N_e values but significantly improved F_{st} estimates. Additionally, the new version can approximate the number of subpopulations involved (see Figures 1, S2, and Table S4). Internally, the approach starts with a two-subpopulation assumption and iteratively corrects for additional subpopulations during parameter estimation.

L. 147: This is very brief and mostly refers to a previous paper and I couldn't follow what exactly the inference steps are. Does this only apply to GONE or to both methods?

RESPONSE:

We apologize for the incorrect citation. To address this, we have extended the explanation (L161 to L178) and corrected the reference to point directly to the specific section in the Appendix where the derivation is detailed.

L. 160: It would be nice to get an overview of how this pseudohaploid version works, here in the main text. I assume you have to do a correction for potentially sampling the other haploid sequence at different sites? Does this also work for currentNE?

RESPONSE:

Row 3 of Figure 2 shows the results of analysing low sequencing depth data, which is equivalent to pseudohaploid data because only one allele from heterozygotes is randomly chosen. The correction for this involves treating the random allele selection as an additional round of free recombination between loci pairs, implemented theoretically as an extra meiosis with free recombination before sampling. This feature is currently implemented only in GONE2 as now explained in the text (L190).

L. 184: Comparing how? How do we know if something is 'significantly different'?

RESPONSE:

This is now better explained in conjunction with the simulation results of Table S4, which show the power of the method to reject panmixia. The interval coefficients of the estimates are used for this purpose.

Figure 2:

- Why does the -x option perform badly under panmixia, for variable N_e ? Shouldn't -x be nested within the default option parameter space? In practice, I will apply both options, notice a difference and then go with -x as suggested which is less accurate.

RESPONSE:

The -x option has the advantage of avoiding the large artefact that occurs because of population structure with low migration, but has the disadvantage of not being precise in showing drastic changes in N_e , given that it gives only linear trends. Our advice is: (1) use the software currentNe2 with and without -x

option to test for population structure. (2) If absence of panmixia is shown then use GONE2 with the `-x` option to get a more reliable trend.

Figure 3:

- I'd appreciate a bit more discussion on Figure 3 in the main text. In my understanding males don't recombine, is that correct? That seems like will have an effect on LD? Can this not explain the patterns you see, of X chromosomes having higher N_e ?

RESPONSE:

Figure 3 is now Figure 4. You are correct that males do not recombine. We account for this by adjusting recombination rates in our analyses. For X chromosomes, the recombination rate is the meiosis rate in females on the typical genetic map. This must be multiplied by $2/3$ (males only carry one copy so $1/3$ of chromosomes do not recombine). For autosomes, the recombination rate must be multiplied by $1/2$ because males do not recombine (males carry two copies, so $2/4$ of the chromosomes do not recombine). Additionally, the haploid N_e estimates for the X chromosomes were rescaled by multiplying by $4/3$, to get the haploid equivalent in autosomes assuming that the number of sexes are equal, and dividing by 2 to get the diploid number. This is now explained in the section Empirical Analyses of the Methods. In reality, the Figure 3 does not represent the N_e for X chromosomes, which is expected to be smaller than the N_e for autosomes. To clarify this, we have added a sentence to the caption of the figure, that was designed to compare sex ratios.

Reviewer #2 (Remarks on code availability):

Code looked good to me! I tried to install it, ran into errors on my Mac. I changed the following to the makefile to get it working:

```
COMMON_FLAGS=-Wall -Xpreprocessor -fopenmp -lomp -  
I/opt/homebrew/Cellar/libomp/19.1.5/include -  
L/opt/homebrew/Cellar/libomp/19.1.5/lib -std=c++11
```

RESPONSE:

Thank you for reviewing the software. It would be good to include these instructions on GitHub. Could you please open an issue on GitHub, or we could consider adding them to the compilation section of the program?

Reviewer #3 (Remarks to the Author):

The authors of this MS describes the extensions made to their previously proposed methodologies for estimating the current and historical effective population sizes N_e from linkage disequilibrium (LD) in marker data. The extensions are made so that the methods apply to subdivided populations for the estimation of both N_e and migration rate (m), and to low quality marker data that either are of low sequencing depth or have genotyping errors. The accuracy and the robustness of these extended methods

are checked by analysing some simulated data, and are also demonstrated by applying to the analysis of drosophila experimental populations with known demography. I believe this research represents significant developments in estimating population demography based on LD in genomic marker data.

RESPONSE:

We thank the reviewer for the comments. They have helped us to improve the tools and the manuscript.

I have just one major and a few minor comments for the authors to consider in revision of this MS.

The major development of this research is to extend the previous LD-based N_e estimation method to apply to subdivided populations. In doing so, it is assumed that a population is subdivided into 2 equal-sized (identical) subpopulations with symmetrical migration rates m (the same m from subpopulation 1 to 2, and from 2 to 1). This simplest subdivision model simplifies the parameter estimation tremendously. However, real populations might be subdivided in many different ways, with a variable number of subpopulations and variable migration rates among subpopulations. These subdivision details do matter with regard to the genetic structure of the population and thus to the N_e of the subdivided population. For example, the number of subpopulations, n , partly determines the N_e . If a population is subdivided into n subpopulations in Wright's (1943) island model, each subpopulation being an idealized population of size N except for receiving a proportion m of immigrants taken randomly from the entire population per generation, the N_e of the population is $N_e = nN(1 + (n-1)^2 / (4Nm))$ derived by Nei & Takahata (1993). Under this model, for a given total population size nN , the higher is the number of subdivisions n , the larger will be the value of N_e . Therefore, assuming $n=2$ in the LD-based method might lead to a biased estimate of N_e .

RESPONSE:

We understand the concerns regarding complex scenarios. To address this we have extended the model to any number of subpopulations. The previous model provided good N_T estimates for two subpopulations but tended to underestimation as the number increased, with less accurate F_{st} estimates. The model now considers the number of subpopulations as an unknown, leading to slightly overestimated N_T values but significantly improved F_{st} estimates (Figures 1, S2, Table S4). The approach starts with a two-subpopulation assumption and iteratively adjusts for additional subpopulations during estimation.

We have also added new sets of simulations (Supplementary Figure S2) which include these scenarios: (A) An island model with differences between the subpopulation sizes; (B) A stepping stone model of migration with different numbers of subpopulations; (C) An island-continent model of migration, where a large population exchanges migrants with different numbers of subpopulations of small size; and (D) An island model with two islands where the migration between them is asymmetrical.

Second, we explain now in the text that the estimate obtained by the software is the total size of the metapopulation (N_T). We agree with the referee in that the metapopulation N_e must consider also the degree of differentiation between populations (F_{st}). Thus, now the software provides the estimate of N_T and that of N_e based on the estimated value of F_{st} . As the reviewer may see in Figures 1 and S2 the estimates of m , F_{st} , N_T and N_e are rather good for a range of circumstances.

L126, the parameter m in the equation is first encountered herein without explanation. What does it represent? Migration rate?

RESPONSE:

Yes, it is now made explicit.

L133, the symbol ΔT^2 is not explained.

RESPONSE:

Corrected. ΔT^2 and Δ^2 were the same thing.

L179, “analysis option ('-x') is used”, where “-x” is not explained. What does it mean?

RESPONSE:

Corrected. This is now made explicit

L279-283, analysing a sample containing individuals sampled from a single subpopulation leads to an estimate of the N_e of the sampled subpopulation, not the entire population. Is this always true, regardless of the value of m ? I think if m is sufficiently large that the subdivided population behaves like a panmictic population, then the LD analysis of a sample of individuals from a single subpopulation should still lead to an estimate of N_e of the entire population.

RESPONSE:

The reviewer is right. If the migration is large, sampling from a single subpopulation will approximate the estimate of the whole metapopulation. This is illustrated, for example, in the paper by Novo et al. (2023a). This is said explicitly in L311.

In Figure 1 and other places, the geometric mean estimates of N_e was used and compared with the simulated value. Why not using arithmetic mean which is commonly used to show the biasness of a method?

RESPONSE:

(this response is common for reviewers 1 and 3)
We chose the geometric mean because the sampling error of linkage disequilibrium (LD) is approximately normally distributed on a log scale. Also, LD is actually the inverse of N_e , and the right way to transform this relationship into a linear function is using a log scale. Transforming to logs, calculating the

arithmetic mean and applying antilogs, is the same as calculating the geometric mean. The geometric mean offers advantages over the arithmetic mean by being less sensitive to outliers and providing a more robust measure of central tendency in skewed distributions.

In Figure 1B, a population of 2000 individuals is subdivided into n equally sized subpopulations with a migration rate $m = 0.001$. The simulated (theoretical) N_e of the population is about 2000, invariable with the value of n (2-7) as shown in 1B. However, under this subdivision model, the simulated (theoretical) N_e values calculated from Nei & Takahata (1993) formula are 2125, 2333, 2562, 2800, 3041 and 3285 for $n=2,3,4,5,6,7$ respectively.

RESPONSE:

Thank you for your insightful comment. In response, we have extended our predictions to include both N_T (the sum of the N_e values of all subpopulations) and N_e . As mentioned in response to the first comment, the previous version of the software only estimated N_T , which was incorrectly referred to as N_e in Figure 1. Figure1 now shows the estimates of N_T and N_e , the latter based on the estimated values of N_T and F_{st} . N_T and N_e are also represented in the new figure S2. The effect that you mentioned is now represented by dashed lines (expectations) and red and blue dots (observations).

In Figure 1D, “Estimates for a metapopulation composed of two unequally sized subpopulations (total 2,000 individuals, $m = 0.001$).” The simulated (theoretical) N_e of the population is about 2000, invariable with the extent of the unbalance in subpopulation size. Is this true?

RESPONSE:

The theory has been developed to predict N_T . The differences in subpopulation sizes do not affect the true metapopulation size in the scenarios of Figure 1D. Consequently, N_T estimates are not expected to change with the imbalance in the distribution of subpopulation sizes, although a slight trend towards decreasing estimates is observed as the imbalance increases. Expectations and estimates of N_T and N_e expectations remain close due to the decrease in F_{st} as most individuals are concentrated in one of the subpopulations. We have added the case 2000/0 (there is only one panmictic population) to Figure 1D. Even in this case the estimate under the incorrect assumption of subdivision (-x option) is about 2000 (slightly above this number) close to the estimate under the correct assumption of panmixia (now commented in line 201), although, the estimate of the number of subpopulations is always 2 (this is a consequence of the assumptions of the model: at least two subpopulations).

RESPONSES TO REVIEWER COMMENTS:

NOTE: The original title (“Advanced Linkage Disequilibrium Methods for Demographic Inference: From Metapopulations to Poor Quality DNA Data”) has been changed to “Accounting for Population Structure and Data Quality in Demographic Inference with Linkage Disequilibrium Methods” to comply with the journal's formatting guidelines.

Reviewer #1 (Remarks to the Author):

Santiago et al present updated methods to infer effective population sizes using linkage disequilibrium. The key innovation of this new work is that they develop approaches that are robust to meta-population structure. This is done by deriving the equations for equilibrium LD under a model of two populations, and then using those in either a moment-based approach currentNe2 or a HMM approach with GONE2. Interestingly, the authors show using simulations that while the methods are developed with two demes in mind, they seem to apply to island models with more demes, and with other violations of the assumptions (such as unequal population or sample sizes). They also show that there is a pretty substantial difference when applied to real data in many cases, with effective sizes being estimated to be much larger when accounting for meta population structure. They also have some additional features to account for sequencing error or low depth of coverage.

RESPONSE:

We thank the reviewer for all his comments. They have helped us to rethink several parts of the theory, the programmes and the manuscript.

This is an interesting and useful contribution to the literature, as certainly many natural populations exist with some form of population structure. The authors simulations show that meta population structure results in under-estimation of effective sizes when there is population structure, in line with the results in real data. The authors suggest that a difference between the population size estimated with and without the assumption of structure can actually be used as a test for population structure. I think that it would enhance the paper to more fully look at the properties of that test: for example, if you reject panmixia when the CIs for Ne between the two methods don't overlap, what are the size and power of the test? I believe the authors can do this using the simulated data they already have, so I do not anticipate it being a major burden, but it will strengthen the paper.

RESPONSE:

Thank you for your insightful comment. We have addressed the suggestion to explore the properties of the test for detecting population structure by comparing the confidence intervals of Ne estimates with and without assuming structure. The additional simulations, detailed in Supplementary Table S4, used an island model with five subpopulations, each with N=400 or N=2000 individuals, and varying migration rates (m). The results show that the method can reject panmixia for $Nm < 10-20$, which is a substantial number of migrants

per generation, indicating that our approach is sensitive to detecting population structure even under moderate to high gene flow.

I think another small and hopefully easily fixable issue with the manuscript is that though the workings of currentNe2 are described in some detail (although I have a couple clarifying questions), I don't believe the workings of GONE2 are described in much detail in the manuscript. I think some description of the algorithm, along the lines of the description of the currentNe2 algorithm, would be helpful, even if it's just in the supplement. There's some description for the GONE2 algorithm in the discussion, which feels like an odd place for it.

RESPONSE:

(this response is common for reviewers 1 and 2)

We have addressed this by providing additional information on how both currentNe2 and GONE2 work to estimate effective population sizes when dealing with metapopulations:

1- We have extended the explanation of the algorithm used to solve the current metapopulation structure with both programs. This includes modifications to Figure S1, additional explanations between lines 119 and 142, and an extension of the "Working with the Software" section in Methods (lines 370-376), which now references a new Figure S5. This figure provides the pseudocode of the algorithm used to determine the unknowns describing a metapopulation..

2- The description of the GONE2 algorithm for estimating temporal effective population sizes in metapopulations has been relocated to lines 143-159, immediately following the introduction of the algorithm for solving current metapopulation structure. This change connects the descriptions of both algorithms, providing a clearer explanation.

Related to both those questions, it doesn't seem that GONE2 has a method of determining whether using the panmictic option or the meta population option is appropriate, as is the case for currentNe2. I feel that some discussion of that point would be appropriate.

RESPONSE:

Unlike currentNe2, GONE2 does not provide a method to decide between models based on confidence intervals. Instead, GONE2 relies on subjective interpretation of the results, such as the proximity of the Ne estimates in the first generation and the presence of a typical drop in historical Ne around generation 5 when using the default option. Both programmes are sensitive to detecting population structure, and combining them is a good option as they are complementary.

I have some more minor comments as well

1) The software described uses only sites on different chromosomes and sites at $c = 0.05$ when they're on the same chromosome. This seems to be throwing out a lot of

data---for example, could currentNe2 use a bunch of different c and a regression/least squares approach to get more out of the data?

RESPONSE:

When the aim is to estimate the current N_e , we require at least two measures of LD that reflect similar drift events, and thus correspond to the same N_e . Most LD between chromosomes ($c=0.5$) or between weakly linked sites ($c>0.05$) reflects very recent drift (L129-139), allowing us to assume that they share the same N_e . While the estimate of LD between chromosomes is a simple average, the LD within chromosome is obtained by integration over all the loci pairs with rec. rates $c>0.05$, giving both points equal weight. We explored alternative approaches before, but the inherent large errors of the measures of LD make them complex and difficult to integrate into the parameter estimation process.

2) GONE2 with errors has to have an error rate pre-specified. In some other contexts (e.g. Racimo et al 2016, Schraiber 2018), it's possible to estimate the error rate. While those are very different models that probably gain identifiability by being "anchored" to "true allele frequencies" in some population, I wonder if there's any hope of estimating an error rate in the GONE2 model? If not, can the authors provide some guidance on how to set the error rate?

RESPONSE:

We believe this is somewhat out of the scope of our software. However, we now cite these methods (L262) to estimate the error rate, which then could be specified when running software.

3) I wonder if, for the real data analysis presented in Table 1, a scatterplot of m vs the log ratio of N_e 's estimated from panmixia vs meta population (i.e. $\log(N_e_{\text{panmixia}}/N_e_{\text{metapop}})$) could be an interesting visualization of that table, to help people understand what's going on.

RESPONSE:

We have created this plot now included as supplementary Figure S4. It shows a linear relationship. The argument explaining this relationship is now included in the manuscript (L243-249).

4) Do the dashed horizontal lines in Table 1 represent some important breaks in the datasets? I couldn't really tell

RESPONSE:

The lines have been removed (they were included only to read more easily the figures across lines).

5) Fig 1: why are simulations summarized as geometric mean instead of arithmetic mean?

RESPONSE:

(this response is common for reviewers 1 and 3)

We chose the geometric mean because the sampling error of linkage disequilibrium (LD) is approximately normally distributed on a log scale. Also, LD is actually the inverse of N_e , and the right way to transform this relationship into a linear function is using a log scale. Transforming to logs, calculating the arithmetic mean and applying antilogs, is the same as calculating the geometric mean. The geometric mean offers advantages over the arithmetic mean by being less sensitive to outliers and providing a more robust measure of central tendency in skewed distributions.

Reviewer #2 (Remarks to the Author):

The authors present new theory and software updates for two previously introduced tools, GONE and currentNe. These tools use LD patterns to estimate very recent effective population sizes. In their updates, both software tools now account for substructures populations and estimate N_e and a migration rate. The paper conducts simulations and applications to real data from *Drosophila* to test the method.

RESPONSE:

We thank the reviewer for all the comments. They have helped us to improve the tools and the manuscript.

These are interesting developments on previously successful tools, both in terms of theory and implementation. However, I found the paper quite hard to read, assuming in many places detailed knowledge about how their previous versions worked.

RESPONSE:

We have rewritten and reorganised the explanations of how both tools work and their connections. You can find these in lines 119 to 159. We have also included Figure S5, which shows the pseudocode of the algorithm used to estimate effective population size in metapopulations. We hope these changes make the paper easier to read.

It is unclear to me how general the proposed model is, and therefore how applicable it is to real scenarios: for instance how does this work if there are multiple subpopulations of different size?

RESPONSE:

Thank you for this comment, which has encouraged us to extend the model so that it can now consider any number of subpopulations. This has now been added to both programs so that an effective number of subpopulations can be estimated. We have also done some new simulations to try out your suggestions. These include an island model with different subpopulation sizes, a stepping stone model with different numbers of subpopulations, an island-continent migration model, and a model with asymmetric migration between two islands. You can see these simulations in Supplementary Figure S2.

As a user, I would like to see more extensive testing on real applications, currently there is only one application of GONE2 and none for currentNe.

RESPONSE:

We give a substantial number of applications of currentNe2 to real data in our Table 1. The table has been improved and updated by using the latest version of the software, and now provides the estimated number of subpopulations (s) and an estimate of the migration rate and the fixation index (F_{st}). In addition, we have now added estimates of historical N_e with GONE2 in Figure 3 for the examples of Table 1. These results are now commented in lines 250 to 257

It would be nice to see the difference in inference between the previous versions and these updates, and comparison with existing recent software tools such as HapLD.

RESPONSE:

The current version of GONE2 gives the same (or very similar) results as the previous version (GONE) if there is not population structure. But the previous version does not account for genotyping errors and low sequencing depth. Likewise, currentNe2 gives the same results as the previous version (currentNe) if there is not population structure. This is now mentioned in the paper to avoid confusions (lines 205 and 217

Regarding HapNe-LD, which is tailored for human data, we have included a comparison with GONE2 in Supplementary Figure S3 and a comment in line 231. This comparison involves simulations mimicking human chromosomes, highlighting the broader applicability of our tools across different species. We appreciate your feedback and hope this addition enhances the paper's interest.

More details on e.g. the number of genomes needed for both tools would be good too.

RESPONSE:

It is a difficult to give a number for a reliable result, which depends on the number of individual and markers in the sample, the population size and the genome size. This problem was addressed in Santiago et al. (2024), resulting in the CIs implemented in currentNe and currentNe2.

L. 35: "metapopulation from a single sample": In my first parse through the paper, I got really confused because this sounded like you can apply the method to a single genome. That seemed strange given you rely on LD. I am still unsure what exactly you mean by sample, but I would definitely consider changing the phrasing here.

RESPONSE:

Thank you for pointing out the potential confusion. Now we make clear that when we say 'single sample' we mean a set of individuals every time we use it.

L. 70: “Infer current N_e from a relatively small number of unphased genotypes”: How many? And compared to what?

RESPONSE:

The sentence is imprecise and a bit unfortunate. As indicated in response to a previous comment, there are several factors affecting accuracy (sample sizes of individuals and polymorphic markers, the true population size and the genome size), which make it impossible to give a number valid in all cases. We have decided to remove the sentence. We think that the range of sample sizes in Table 1 gives an indication of the applicability of the method.

L. 98: Here I would add at least a section on how GONE and current N_e work. I had to go back to your other papers.

RESPONSE:

(this response is common for reviewers 1 and 2)

We have addressed this by providing additional information on how both current N_e 2 and GONE2 work to estimate effective population sizes when dealing with metapopulations:

1- We have expanded the explanation of the algorithm used to solve the current metapopulation structure with both programs. This includes modifications to Figure S1, additional explanations between lines 119 and 142, and an extension of the "Working with the Software" section in Methods (lines 370-375), which now references a new Figure S5. This figure provides the pseudocode of the algorithm used to determine the unknowns describing a metapopulation..

2- The description of the GONE2 algorithm for estimating temporal effective population sizes in metapopulations has been relocated to lines 143-159, immediately following the introduction of the algorithm for solving current metapopulation structure. This change connects the descriptions of both algorithms, providing a clearer explanation.

L. 109: I am confused by the statement: “we show that a two-population model with reciprocal migration rate m can effectively approximate the effective size of the entire metapopulation using a single sample”: Is the claim that even if you have more than 2 populations in reality, your approach infers N_e and migration rates correctly assuming only 2 populations? That seems quite strange to me, and would also presumably depend on how different migration rates are between different populations, differences in their N_e etc?

RESPONSE:

Assuming two subpopulations gave quite good estimates of N_e when the true subpopulation number was larger, although estimates progressively deviated to underestimation as the number increases. The good prediction is partly due to fact that the model was built to predict N_T , regardless how individuals are distributed among subpopulations. However, the F_{st} estimates were less

accurate. The updated model now considers any number of subpopulations, leading to slightly overestimated N_e values but significantly improved F_{st} estimates. Additionally, the new version can approximate the number of subpopulations involved (see Figures 1, S2, and Table S4). Internally, the approach starts with a two-subpopulation assumption and iteratively corrects for additional subpopulations during parameter estimation.

L. 147: This is very brief and mostly refers to a previous paper and I couldn't follow what exactly the inference steps are. Does this only apply to GONE or to both methods?

RESPONSE:

We apologize for the incorrect citation. To address this, we have extended the explanation (L161 to L178) and corrected the reference to point directly to the specific section in the Appendix where the derivation is detailed.

L. 160: It would be nice to get an overview of how this pseudohaploid version works, here in the main text. I assume you have to do a correction for potentially sampling the other haploid sequence at different sites? Does this also work for currentNE?

RESPONSE:

Row 3 of Figure 2 shows the results of analysing low sequencing depth data, which is equivalent to pseudohaploid data because only one allele from heterozygotes is randomly chosen. The correction for this involves treating the random allele selection as an additional round of free recombination between loci pairs, implemented theoretically as an extra meiosis with free recombination before sampling. This feature is currently implemented only in GONE2 as now explained in the text (L190).

L. 184: Comparing how? How do we know if something is 'significantly different'?

RESPONSE:

This is now better explained in conjunction with the simulation results of Table S4, which show the power of the method to reject panmixia. The interval coefficients of the estimates are used for this purpose.

Figure 2:

- Why does the -x option perform badly under panmixia, for variable N_e ? Shouldn't -x be nested within the default option parameter space? In practice, I will apply both options, notice a difference and then go with -x as suggested which is less accurate.

RESPONSE:

The -x option has the advantage of avoiding the large artefact that occurs because of population structure with low migration, but has the disadvantage of not being precise in showing drastic changes in N_e , given that it gives only linear trends. Our advice is: (1) use the software currentNe2 with and without -x

option to test for population structure. (2) If absence of panmixia is shown then use GONE2 with the `-x` option to get a more reliable trend.

Figure 3:

- I'd appreciate a bit more discussion on Figure 3 in the main text. In my understanding males don't recombine, is that correct? That seems like will have an effect on LD? Can this not explain the patterns you see, of X chromosomes having higher N_e ?

RESPONSE:

Figure 3 is now Figure 4. You are correct that males do not recombine. We account for this by adjusting recombination rates in our analyses. For X chromosomes, the recombination rate is the meiosis rate in females on the typical genetic map. This must be multiplied by $2/3$ (males only carry one copy so $1/3$ of chromosomes do not recombine). For autosomes, the recombination rate must be multiplied by $1/2$ because males do not recombine (males carry two copies, so $2/4$ of the chromosomes do not recombine). Additionally, the haploid N_e estimates for the X chromosomes were rescaled by multiplying by $4/3$, to get the haploid equivalent in autosomes assuming that the number of sexes are equal, and dividing by 2 to get the diploid number. This is now explained in the section Empirical Analyses of the Methods. In reality, the Figure 3 does not represent the N_e for X chromosomes, which is expected to be smaller than the N_e for autosomes. To clarify this, we have added a sentence to the caption of the figure, that was designed to compare sex ratios.

Reviewer #2 (Remarks on code availability):

Code looked good to me! I tried to install it, ran into errors on my Mac. I changed the following to the makefile to get it working:

```
COMMON_FLAGS=-Wall -Xpreprocessor -fopenmp -lomp -  
I/opt/homebrew/Cellar/libomp/19.1.5/include -  
L/opt/homebrew/Cellar/libomp/19.1.5/lib -std=c++11
```

RESPONSE:

Thank you for reviewing the software. It would be good to include these instructions on GitHub. Could you please open an issue on GitHub, or we could consider adding them to the compilation section of the program?

Reviewer #3 (Remarks to the Author):

The authors of this MS describes the extensions made to their previously proposed methodologies for estimating the current and historical effective population sizes N_e from linkage disequilibrium (LD) in marker data. The extensions are made so that the methods apply to subdivided populations for the estimation of both N_e and migration rate (m), and to low quality marker data that either are of low sequencing depth or have genotyping errors. The accuracy and the robustness of these extended methods

are checked by analysing some simulated data, and are also demonstrated by applying to the analysis of drosophila experimental populations with known demography. I believe this research represents significant developments in estimating population demography based on LD in genomic marker data.

RESPONSE:

We thank the reviewer for the comments. They have helped us to improve the tools and the manuscript.

I have just one major and a few minor comments for the authors to consider in revision of this MS.

The major development of this research is to extend the previous LD-based N_e estimation method to apply to subdivided populations. In doing so, it is assumed that a population is subdivided into 2 equal-sized (identical) subpopulations with symmetrical migration rates m (the same m from subpopulation 1 to 2, and from 2 to 1). This simplest subdivision model simplifies the parameter estimation tremendously. However, real populations might be subdivided in many different ways, with a variable number of subpopulations and variable migration rates among subpopulations. These subdivision details do matter with regard to the genetic structure of the population and thus to the N_e of the subdivided population. For example, the number of subpopulations, n , partly determines the N_e . If a population is subdivided into n subpopulations in Wright's (1943) island model, each subpopulation being an idealized population of size N except for receiving a proportion m of immigrants taken randomly from the entire population per generation, the N_e of the population is $N_e = nN(1 + (n-1)^2 / (4Nm))$ derived by Nei & Takahata (1993). Under this model, for a given total population size nN , the higher is the number of subdivisions n , the larger will be the value of N_e . Therefore, assuming $n=2$ in the LD-based method might lead to a biased estimate of N_e .

RESPONSE:

We understand the concerns regarding complex scenarios. To address this we have extended the model to any number of subpopulations. The previous model provided good N_T estimates for two subpopulations but tended to underestimation as the number increased, with less accurate F_{st} estimates. The model now considers the number of subpopulations as an unknown, leading to slightly overestimated N_T values but significantly improved F_{st} estimates (Figures 1, S2, Table S4). The approach starts with a two-subpopulation assumption and iteratively adjusts for additional subpopulations during estimation.

We have also added new sets of simulations (Supplementary Figure S2) which include these scenarios: (A) An island model with differences between the subpopulation sizes; (B) A stepping stone model of migration with different numbers of subpopulations; (C) An island-continent model of migration, where a large population exchanges migrants with different numbers of subpopulations of small size; and (D) An island model with two islands where the migration between them is asymmetrical.

Second, we explain now in the text that the estimate obtained by the software is the total size of the metapopulation (N_T). We agree with the referee in that the metapopulation N_e must consider also the degree of differentiation between populations (F_{st}). Thus, now the software provides the estimate of N_T and that of N_e based on the estimated value of F_{st} . As the reviewer may see in Figures 1 and S2 the estimates of m , F_{st} , N_T and N_e are rather good for a range of circumstances.

L126, the parameter m in the equation is first encountered herein without explanation. What does it represent? Migration rate?

RESPONSE:

Yes, it is now made explicit.

L133, the symbol ΔT^2 is not explained.

RESPONSE:

Corrected. ΔT^2 and Δ^2 were the same thing.

L179, “analysis option ('-x') is used”, where “-x” is not explained. What does it mean?

RESPONSE:

Corrected. This is now made explicit

L279-283, analysing a sample containing individuals sampled from a single subpopulation leads to an estimate of the N_e of the sampled subpopulation, not the entire population. Is this always true, regardless of the value of m ? I think if m is sufficiently large that the subdivided population behaves like a panmictic population, then the LD analysis of a sample of individuals from a single subpopulation should still lead to an estimate of N_e of the entire population.

RESPONSE:

The reviewer is right. If the migration is large, sampling from a single subpopulation will approximate the estimate of the whole metapopulation. This is illustrated, for example, in the paper by Novo et al. (2023a). This is said explicitly in L311.

In Figure 1 and other places, the geometric mean estimates of N_e was used and compared with the simulated value. Why not using arithmetic mean which is commonly used to show the biasness of a method?

RESPONSE:

(this response is common for reviewers 1 and 3)
We chose the geometric mean because the sampling error of linkage disequilibrium (LD) is approximately normally distributed on a log scale. Also, LD is actually the inverse of N_e , and the right way to transform this relationship into a linear function is using a log scale. Transforming to logs, calculating the

arithmetic mean and applying antilogs, is the same as calculating the geometric mean. The geometric mean offers advantages over the arithmetic mean by being less sensitive to outliers and providing a more robust measure of central tendency in skewed distributions.

In Figure 1B, a population of 2000 individuals is subdivided into n equally sized subpopulations with a migration rate $m = 0.001$. The simulated (theoretical) N_e of the population is about 2000, invariable with the value of n (2-7) as shown in 1B. However, under this subdivision model, the simulated (theoretical) N_e values calculated from Nei & Takahata (1993) formula are 2125, 2333, 2562, 2800, 3041 and 3285 for $n=2,3,4,5,6,7$ respectively.

RESPONSE:

Thank you for your insightful comment. In response, we have extended our predictions to include both N_T (the sum of the N_e values of all subpopulations) and N_e . As mentioned in response to the first comment, the previous version of the software only estimated N_T , which was incorrectly referred to as N_e in Figure 1. Figure1 now shows the estimates of N_T and N_e , the latter based on the estimated values of N_T and F_{st} . N_T and N_e are also represented in the new figure S2. The effect that you mentioned is now represented by dashed lines (expectations) and red and blue dots (observations).

In Figure 1D, “Estimates for a metapopulation composed of two unequally sized subpopulations (total 2,000 individuals, $m = 0.001$).” The simulated (theoretical) N_e of the population is about 2000, invariable with the extent of the unbalance in subpopulation size. Is this true?

RESPONSE:

The theory has been developed to predict N_T . The differences in subpopulation sizes do not affect the true metapopulation size in the scenarios of Figure 1D. Consequently, N_T estimates are not expected to change with the imbalance in the distribution of subpopulation sizes, although a slight trend towards decreasing estimates is observed as the imbalance increases. Expectations and estimates of N_T and N_e expectations remain close due to the decrease in F_{st} as most individuals are concentrated in one of the subpopulations. We have added the case 2000/0 (there is only one panmictic population) to Figure 1D. Even in this case the estimate under the incorrect assumption of subdivision (-x option) is about 2000 (slightly above this number) close to the estimate under the correct assumption of panmixia (now commented in line 201), although, the estimate of the number of subpopulations is always 2 (this is a consequence of the assumptions of the model: at least two subpopulations).